# Post-synthetic modification of covalent organic frameworks for CO$_2$ electroreduction

Minghao Liu[1,2,8], Shuai Yang[1,3,8], Xiubei Yang[1,4], Cheng-Xing Cui [5,6] ✉, Guojuan Liu[1,4], Xuewen Li[1,4], Jun He [2], George Zheng Chen [7], Qing Xu [1,4] ✉ & Gaofeng Zeng [1,4] ✉

To achieve high-efficiency catalysts for CO$_2$ reduction reaction, various catalytic metal centres and linker molecules have been assembled into covalent organic frameworks. The amine-linkages enhance the binding ability of CO$_2$ molecules, and the ionic frameworks enable to improve the electronic conductivity and the charge transfer along the frameworks. However, directly synthesis of covalent organic frameworks with amine-linkages and ionic frameworks is hardly achieved due to the electrostatic repulsion and predicament for the strength of the linkage. Herein, we demonstrate covalent organic frameworks for CO$_2$ reduction reaction by modulating the linkers and linkages of the template covalent organic framework to build the correlation between the catalytic performance and the structures of covalent organic frameworks. Through the double modifications, the CO$_2$ binding ability and the electronic states are well tuned, resulting in controllable activity and selectivity for CO$_2$ reduction reaction. Notably, the dual-functional covalent organic framework achieves high selectivity with a maximum CO Faradaic efficiency of 97.32% and the turnover frequencies value of 9922.68 h$^{-1}$, which are higher than those of the base covalent organic framework and the single-modified covalent organic frameworks. Moreover, the theoretical calculations further reveal that the higher activity is attributed to the easier formation of immediate *CO from COOH*. This study provides insights into developing covalent organic frameworks for CO$_2$ reduction reaction.

Covalent organic frameworks (COFs) are fully designable porous polymers, being comprised of organic knots and covalently connected linkers[1–4]. The diversity of their building units allows them to have a high degree of structural tenability in terms of skeletons, porosities and topologies, resulting in tailorable functions[5–8]. The functional COFs have been utilized in molecular absorptions, light emitters, photo conduction/catalysis, proton/lithium-ion conduction, and lithium-ion or lithium-sulfur batteries[9–16]. Notably, the stable frameworks, large surface areas, open porous channels, and predictable catalytic sites of COFs also allow them to be used as electrocatalysts in the electrochemical reactions, such as oxygen reduction reaction, oxygen evolution reaction, hydrogen evolution reaction, CO$_2$ reduction reactions (CO$_2$RR), and H$_2$O$_2$ synthesis[17–22]. Electrocatalytic CO$_2$RR, which can produce high-value products from the greenhouse gas, is a promising strategy for addressing the CO$_2$ crisis. Owing to the molecular designability, COFs are an ideal class of templates to construct catalysts for CO$_2$RR[23–26]. The catalytic behavior of COFs for CO$_2$RR is dependent on the structure and properties[27–30]. From the

perspective of structure, the types of catalytic centers, electronic states of the linker molecules, and linkage diversities have been adopted to assemble different catalytic COFs[31–34]. By changing the building units or linkages, the corresponding properties of COFs, such as the binding ability of $CO_2$ and the electron conductivity, can be adjusted[35–37]. Thus, a COF structure containing both amine bonds and ionizing skeletons, which enhance the binding ability of $CO_2$ molecules and the framework conductivity, respectively, is desirable for highly efficient $CO_2RR$.

The bottom-up synthetic approach is the most common method to directly construct functional COFs[38–42]. Challenged by the steric hindrance effect, the solubility difference and the microscopic reversibility, however, some functionalities cannot be introduced directly into COFs via the bottom-up syntheses[43–47]. For the pre-designed COF that contains amine bonds and ionizing skeletons, the existing covalent connection methods (e.g., boroxine rings[48], imine bonds[49] and β-ketoenamine linkage[50]) do not support to obtain C−N linkage through the bottom-up synthesis directly, and the electrostatic repulsion effects of ionic building units impede the direct formation of ionizing skeletons. Alternatively, the post-synthetic modification strategy provides a promising chance to construct functional skeletons, pores, and linkages at the molecular level with controllable catalytic properties[51–56]. Several single-step post-modifications were proposed to endow the base COFs with special properties[57–60]. Deng et al. constructed imine-COFs by a post-reduction method to improve gas diffusion on electrode[61]. Guo et al. used Viologens $(C_5H_4NR)_2^{n+}$ to construct a cationic radical framework from the 2,2′-bipyridine-based COF, which showed high photothermal conversion efficiencies[62]. Thus, it is expectable to obtain the COF with reduced imine linkages and ionic skeletons if the single-step post-modification of reduction and ionization can be well integrated. However, the integrated multilevel post-modification is rarely reported, because that not only a robust base COF is required but also the interference of multistep should be well avoided. In the view of COF design, 4,4′,4″,4‴-(1,4-phenylenebis(azanetriyl))tetra-benzaldehyde (PATA) unit contains ammonium groups which are potential to transfer to ionic skeleton, and 5,10,15,20-tetrakis(4-aminophenyl)porphinato]-cobalt (TAPP(Co)) unit can achieve efficient charge transfer due to their conjugated macrocyclic structures. In addition, we previously proved that CoTAPP-PATA-COF, composed by PATA and TAPP(Co), possesses good crystallinity, high surface areas, and excellent chemical stability[28]. Thus, CoTAPP-PATA-COF is expected as a template for constructing multilevel post-synthetic modification COFs for $CO_2RR$.

In this study, we demonstrate a multilevel post-synthetic modification strategy to construct catalytic COFs for $CO_2RR$. The N⁺-NH-COF (N⁺: ionic modification; NH: reduction modification), which was constructed from CoTAPP-PATA-COF by the multilevel post-synthetic modification, showed a maximum CO faradic efficiency of 97.32% at −0.8 V with a CO current density of −28.01 mA cm⁻² and the TOF value of 9922.68 h⁻¹ at −1.0 V vs. RHE. The excellent electrocatalytic properties can be attributed to the superior binding ability of $CO_2$ molecules from C−N bonds and high conductivity from ionization skeletons in these COFs. The results showed that reduction of C=N linkages into the C−N bond and ionization of the linkers significantly improve the selectivity and activity.

## Results

### Chemistry and structure of N⁺-NH-COF

The base COF (CoTAPP-PATA-COF) was synthesized from CoTAPP and PATA using the solvothermal method described in our previous study[17]. The C=N linkages of the COF were reduced by adding NaBH₄ in dimethylacetamide (DMAC) to yield NH-COF, and the PATA units were ionized through in situ ammonium groups via Menshutkin reactions to obtain the N⁺-COF (Fig. 1a)[63]. The base COF was modified through

sequential double post-functionalization (reduction and ionization) to obtain the N⁺-NH-COF (Fig. 1b).

The successful functionalization on the linkage reduction and the skeleton ionization were elucidated by the Fourier transform infrared spectroscopy (FTIR). Compared with the base COF, the C=N linkages at 1622 cm⁻¹ were retained, while a peak raised at 1470 cm⁻¹, being ascribed to -N⁺-(CH₃)₂⁻, in the FTIR spectrum of ionized N⁺-COF (Supplementary Fig. 1)[31,56]. This reveals that the skeleton ionization was solely realized by the Menshutkin reaction. After the reduction modification, in contrast, the FTIR spectra of NH-COF and the N⁺-NH-COF revealed that the C=N vibrations at 1622 cm⁻¹ were totally replaced by the signals of C−N at 1157 cm⁻¹ (Supplementary Fig. 1)[61]. This indicates that the C=N linkages were fully converted into C−N bonds. Apart from C−N vibrations, furthermore, the ionic -N⁺-(CH₃)₂⁻ was also detected for the FTIR spectrum of N⁺-NH-COF. Moreover, the Co−N bonds at 660 cm⁻¹ were observed in all four COFs, indicating that the post-modifications were harmless to the Co-N coordinations.

The functionalizations of COFs were further confirmed by using the solid ¹³C cross-polarization/magic-angle spinning solid-state nuclear magnetic resonance (CP/MAS ssNMR, Supplementary Fig. 2). The C=N signals (148 ppm) in CoTAPP-PATA-COF and N⁺-COF were completely replaced by the C−N signals at 51 ppm for the NH-COF and N⁺-NH-COF. In addition, N⁺-COF and N⁺-NH-COF exhibited a peak belonging to methyl groups at 60 ppm. These results indicate that the successful reduction of −C=N and the methylation with charge state changes. On the other hand, the FTIR and ¹³C NMR results also reveal that the Menshutkin reaction is selective for the methylzation of C−N bonds rather than the co-exsited C=N bonds[60].

The crystalline structures of CoTAPP-PATA-COF, N⁺-COF, NH-COF and N⁺-NH-COF were investigated by the powder X-ray diffraction (PXRD) measurements. The PXRD pattern of base CoTAPP-PATA-COF showed the peaks of (011), (022), (031) and (001) facets at 5.15, 11.02, 12.23 and 21.84 °, respectively (Fig. 2a). The Pawley refinements revealed that the theoretical structures were in accordance with the experimental results with $R_{wp}$ and $R_p$ of 3.04% and 2.98%, respectively (Supplementary Table 1). According to the self-consistent charge density functional tight binding (DFTB) method, the base COF adopted AA stacking model, which enables to provide the open channels for mass transport (Fig. 2a). For the NH-COF, the peaks from (011), (310) and (001) were also identified (Fig. 2b). The Pawley refinements revealed that the simulated results were in accordance with the experimental results. And NH-COF adopts an eclipsed stacking in a PM space group with refined cell parameters of $a = 22.51$ Å, $b = 23.45$ Å, $c = 5.41$ Å, $\alpha = \beta = \gamma = 90$ °, with the corresponding $R_{wp}$ of 2.64% and $R_p$ of 2.06% (Supplementary Table 2). The N⁺-COF also displayed a good crystallinity with intense peaks at 5.15° (011), 11.12° (022), and 21.97° (001) (Fig. 2c). Both of the ionized COFs adopted AA stacking models. Notably, the crystallinity of the N⁺-NH-COF was well maintained after the double post-modifications, offering the peaks of (011), (002), (202) and (001) at 5.15, 11.12, 12.32 and 22.17 °, respectively (Fig. 2d). The highly crystalline structures were further confirmed by the Pawley refinements, delivering the corresponding $R_{wp}$ of 2.67% and 3.78%, $R_p$ of 2.26% and 3.03% for N⁺-COF and N⁺-NH-COF, respectively (Supplementary Tables 3, 4). Furthermore, the Pawley-refined patterns of four COFs illustrated that the AB stacking simulated structures were different to the experimental results (Supplementary Figs. 3–6).

The porous structures of COFs are crucial for the mass transport and the accessibility of active sites during electrocatalysis. Thus, the porosity of these COFs was investigated through the nitrogen sorption isotherm measurements at 77 K. The CoTAPP-PATA-COF exhibited a microporous sorption behaviour with a Brunauer−Emmett−Teller (BET) surface area ($S_{BET}$) of 943.72 m² g⁻¹ (Fig. 3a, black curve), delivering a pore volume of 0.75 cm³ g⁻¹ and a pore size of 1.1 nm (Supplementary Fig. 7). After the post modifications, the $S_{BET}$ changed to 410.69, 659.05, and 340.19 m² g⁻¹ for N⁺-COF, NH-COF, and N⁺-NH-COF,

respectively. In addition, the corresponding pore volumes declined to 0.45, 0.58, and 0.46 $cm^3 g^{-1}$, while the pore sizes kept no change for $N^+$-COF, NH-COF, and $N^+$-NH-COF, respectively (Supplementary Fig. 7). Considering the critical roles of $CO_2$ absorption on $CO_2$RR, we investigated the $CO_2$ sorption behaviours at 273 K. The $CO_2$ uptakes of the CoTAPP-PATA-COF, $N^+$-COF, NH-COF, and $N^+$-NH-COF were 46.43, 27.80, 39.23, and 22.50 $cm^3 g^{-1}$ at 1 bar, respectively (Fig. 3b). Benefiting from the abundant C−N bonds, the NH-COF showed a high $CO_2$ adsorption capacity although its surface area and pore volume were relatively decreased. The role of C−N bonds on $CO_2$ adsorption was further checked by the $CO_2$-temperature programmed desorption ($CO_2$-TPD). Specifically, NH-COF exhibited a higher $CO_2$ desorption signal than that of base CoTAPP-PATA-COF, indicating the enhanced $CO_2$ adsorption capacity on C−N bonds (Supplementary Fig. 8).

The morphologies of the COFs were studied via the field-emission scanning electron microscopy (FE-SEM) and transmission electron microscopy (TEM). The post-modified COFs exhibited similar morphologies to that of base COF without significant morphological changes, suggesting that the structure is well preserved (Supplementary Figs. 9–12). This was further confirmed by the TEM observations (Supplementary Figs. 13–16). The high-resolution TEM (HR-TEM) images showed the ordered straight channels with diameters of ~1.1 nm in the COFs, in accordance with the pore sizes determined by the $N_2$ absorptions (Supplementary Fig. 17). It indicates that the mass transfer paths and the crystallinity of post-modified COFs were well protected. Energy dispersive X-ray spectroscopy (EDX) mapping images revealed that all elements were uniformly distributed in the COFs (Supplementary Figs. 18–21). Furthermore, the thermal gravimetric analysis (TGA) measurements showed that the post-modified COFs kept the similar thermal stability to that of base COF, and no significant mass loss was observed at the temperature <480 °C in $N_2$ (Supplementary Fig. 22).

In addition, the hydrophobicity of these COFs was evaluated by the water contact angle (WCA) measurements (Supplementary Fig. 23). The measured WCAs of the CoTAPP-PATA-COF, $N^+$-COF, NH-COF, and $N^+$-NH-COF were 121.6 ± 3.2 °, 127.1 ± 2.6 °, 129.4 ± 3.1 °, and 128.3 ± 2.7 °, respectively, indicating that the effects of post modifications on the surface hydrophilicity are negligible. Moreover, the hydrophobic behaviour of $N^+$-NH-COF was a further verified by water uptake test. The $N^+$-NH-COF exhibited a low water vapour uptake capacity of 188 $cm^3 g^{-1}$ at 298 K and a inflection point at high relative pressures (P/P$_0$ = 0.5), which are in good agreement with the hydrophobic behaviours (Supplementary Fig. 24)[64]. For the electrocatalysis of $CO_2$RR, the hydrophobic nature of catalyst is crucial to protect the active sites through suppressing the competitive adsorption of water, leading to the improvements of selectivity and energy efficiency[65].

The chemical structures and electron states of the COFs were investigated via X-ray photoelectron spectroscopy (XPS). The XPS spectra showed peaks corresponding to C, N, O, and Co in the prepared COFs (Supplementary Fig. 25). The Co content was 4.0, 4.0, 3.7, and 3.7 wt.% in the CoTAPP-PATA-COF, $N^+$-COF, NH-COF, and $N^+$-NH-COF, respectively, which are close to the values obtained from the inductively coupled plasma (ICP) measurements (i.e., 3.6, 3.5, 3.3, and 3.1wt.%, respectively). Furthermore, the high-resolution Co 2p spectra of the four COFs exhibited Co-N coordination, which confirmed that the Co-N sites were well retained after the multilevel postmodifications (Fig. 4a). In detail, compared with CoTAPP-PATA-COF (781.38 eV) and NH-COF (781.36 eV), the Co 2$p_{3/2}$ spectra of $N^+$-COF (780.92 eV) and $N^+$-NH-COF (780.90 eV) showed a negative shift of ~0.45 eV, which is ascribed to the electron-withdrawing effect of

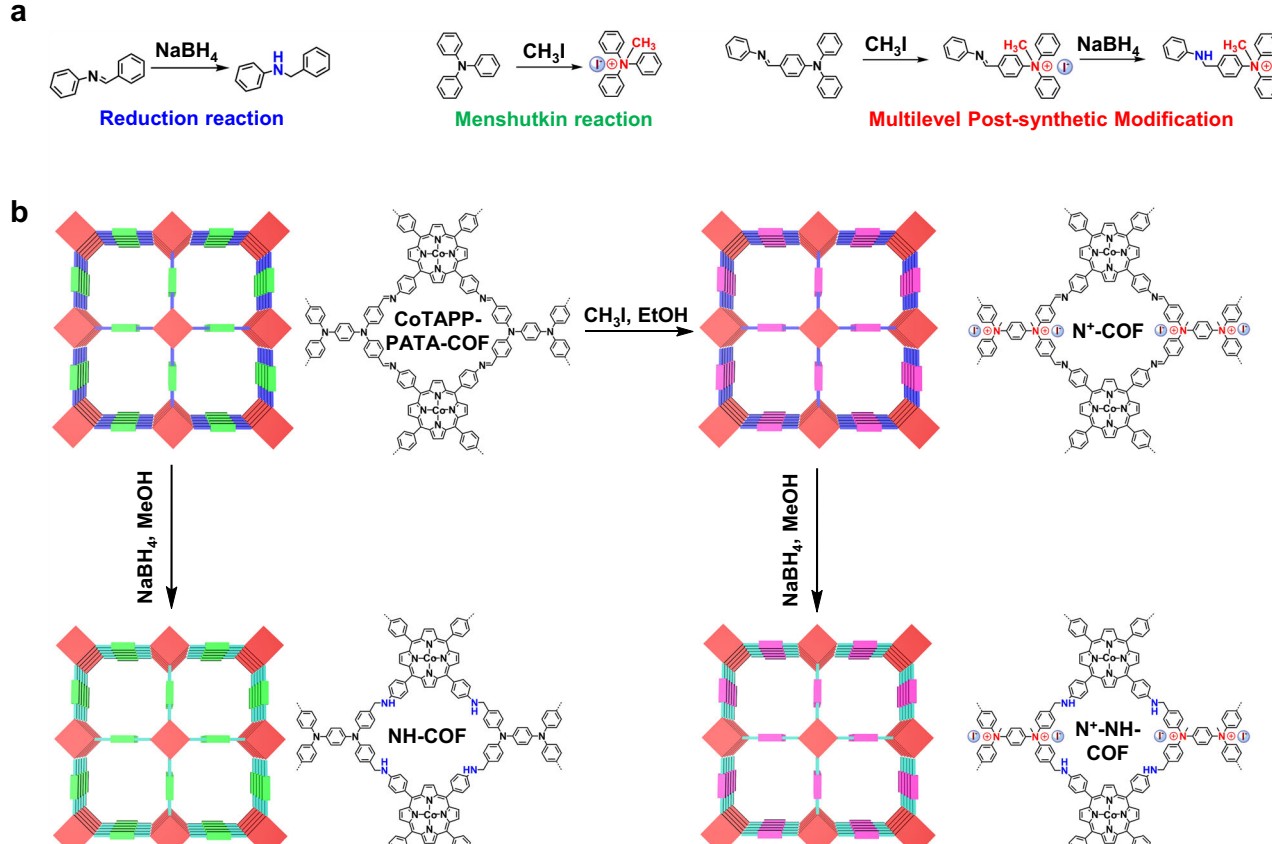

**Fig. 1 | Schematic illustration of multilevel post-synthetic modification. a** Effects of reduction reaction, Menshutkin reaction and multilevel post-synthetic modification on the bond change and charge state. **b** The synthesis of $N^+$-COF, NH-COF and $N^+$-NH-COF from the base COF (CoTAPP-PATA-COF).

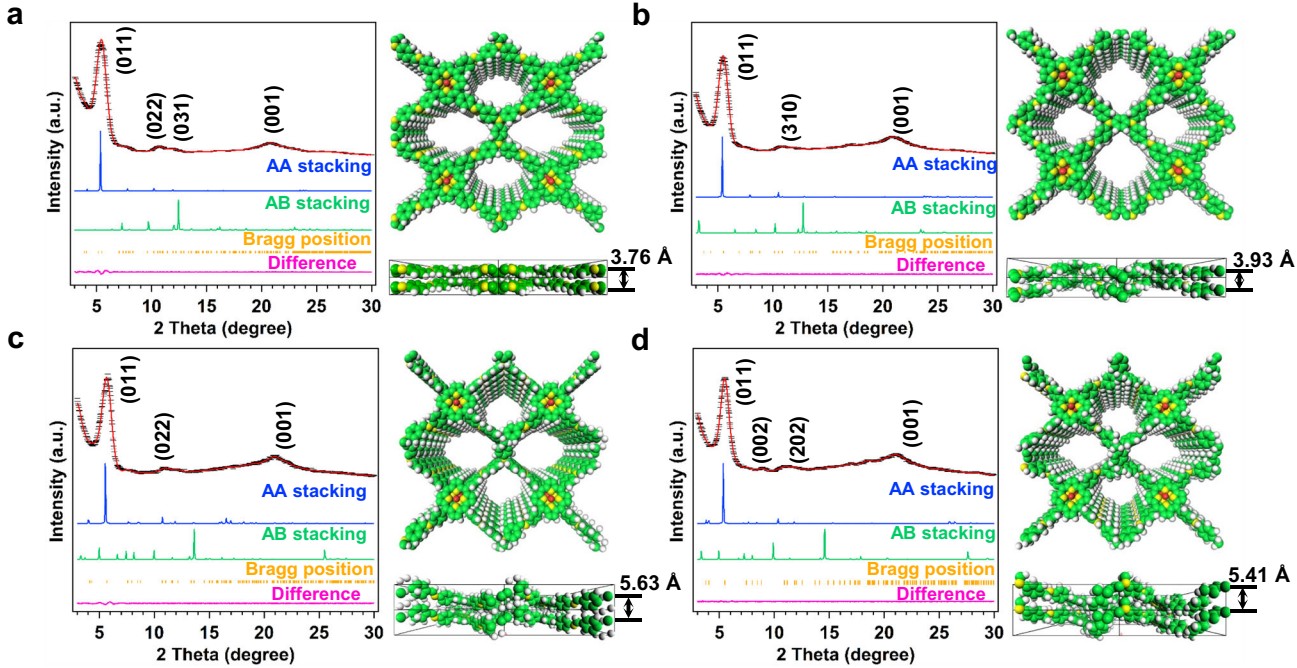

**Fig. 2 | PXRD patterns of four COFs.** The PXRD profiles and simulated structures of **a** CoTAPP-PATA-COF, **b** NH-COF, **c** N$^+$-COF and **d** N$^+$-NH-COF. Line colour follows: experimentally observed (black), Pawley refined (red), Bragg positions (orange) and their difference (pink), simulated using the AA (blue) and staggered AB (green) stacking modes. Atom colour in the theoretically modelled eclipsed-AA stacking models: C-green, N-yellow, H-white, Co-red.

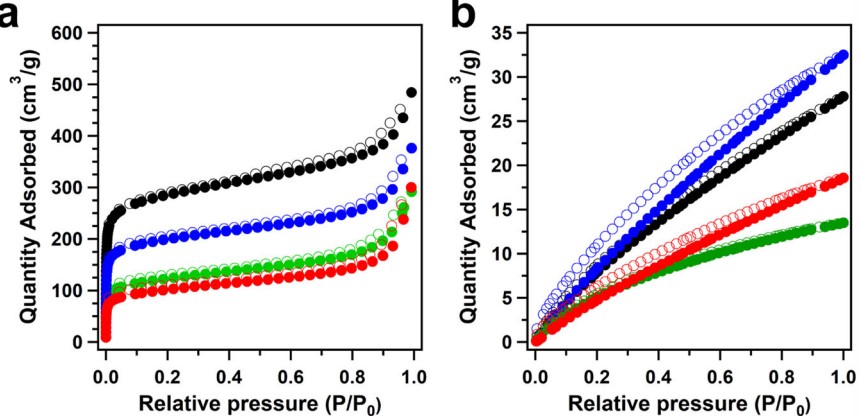

**Fig. 3 | Gas adsorption characterization of four COFs. a** The nitrogen-sorption isotherms at 77 K and **b** the CO$_2$ absorption curves at 273 K for CoTAPP-PATA-COF (black), N$^+$-COF (green), NH-COF (blue) and N$^+$-NH-COF (red).

methyl groups[56]. This proves that the skeletons in N$^+$-COF and N$^+$-NH-COF have been ionized by the CH$_3$I modification. The high-resolution $N$ 1 s spectrum of the CoTAPP-PATA-COF was deconvoluted into three peaks of pyrrole N (398.29 eV), C=N (399.62 eV) and C−N (401.03 eV) with relative contents of 28.14, 55.88 and 15.97 at.%, respectively (Fig. 4b). In comparison, the $N$ 1 s spectrum of the NH-COF showed that the content of C=N was decreased to 14.74% while a C-NH peak appeared at 400.23 eV with a content of 46.09%. Moreover, the N$^+$-COF displayed a peak at 402.34 eV, which was attributed to N$^+$-CH$_3$ bonds[63]. Due to the reduced linkages and the ionic linkers, the peaks of C-NH and N$^+$-CH$_3$ were identified at 400.03 and 402.34 eV, respectively, in the N 1 s spectrum of the N$^+$-NH-COF (Fig. 4b). Therefore, the linkages and skeletons were fully modulated, as expected.

To investigate the different properties of the base and functionalized COFs, the ultraviolet−visible (UV-Vis) spectroscopy was adopted to determine their band gaps (Fig. 4c). Accordingly, the band gaps of the CoTAPP-PATA-COF, N$^+$-COF, NH-COF, and N$^+$-NH-COF were determined as 2.68, 2.52, 2.66, and 2.67 eV, respectively (Fig. 4c, inset). As the conductivity increased with the decrease of gap value, the N$^+$-COF has the highest conductivity compared with that of other COFs, indicating that the ionization units could improve the conductivity. To confirm the electronic conductivity change, the CoTAPP-PATA-COF, N$^+$-COF, NH-COF and N$^+$-NH-COF were measured by the four-probe method at 298 K (Supplementary Fig. 26). N$^+$-NH-COF and N$^+$-COF have similar electronic conductivities at $6.7 \times 10^{-9}$ S m$^{-1}$ and $8.1 \times 10^{-9}$ S m$^{-1}$, respectively, which are one order of magnitude larger than those of CoTAPP-PATA-COF ($8.5 \times 10^{-10}$ S m$^{-1}$) and NH-COF ($3.0 \times 10^{-10}$ S m$^{-1}$). It suggests that the ionized skeletons can promote the electron transfer along the frameworks, and thus improving the activity. The highest occupied molecular orbital (HOMO) and lowest unoccupied molecular orbital (LUMO) were calculated to investigate the electron conduction properties using the Mott−Schottky method (Supplementary Fig. 27).

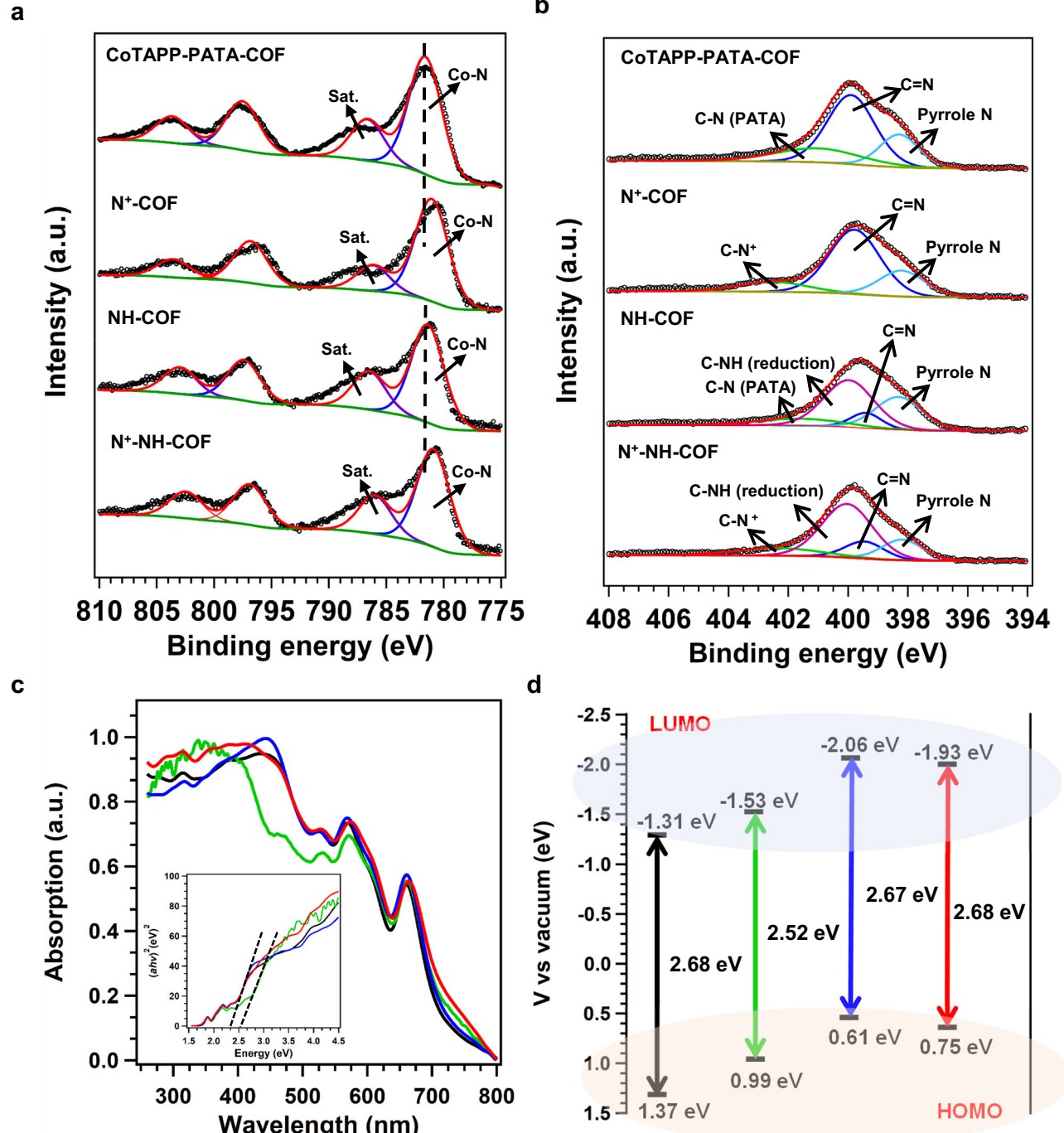

**Fig. 4 | Electronic characterizations of four COFs.** The XPS spectra of **a** Co *2p* and **b** N *1s* for CoTAPP-PATA-COF, N⁺-COF, NH-COF and N⁺-NH-COF. **c** The UV-vis absorption (insert: Tauc plots) and **d** the energy gap (HOMO and LUMO) for CoTAPP-PATA-COF (black), N⁺-COF (green), NH-COF (blue) and N⁺-NH-COF (red).

The HOMO positions of the CoTAPP-PATA-COF, N⁺-COF, NH-COF, and N⁺-NH-COF were 1.37, 0.99, 0.61, and 0.75 eV, respectively, suggesting that the multilevel post-synthetic modification effectively improved the reduction capacity of COFs (Fig. 4d)[58].

## CO₂RR performance on N⁺-NH-COF

The CO₂RR performance of the catalytic COFs was investigated in a KHCO₃ aqueous solution (0.5 M, pH 7.2) saturated with CO₂ using a standard two-compartment electrochemical cell. The COFs were mixed with carbon black at a weight ratio of 5/8. Firstly, the linear sweep voltammetry (LSV) measurements were conducted at a scan rate of 10 mV s⁻¹ from 0 to −1.0 V vs. RHE (Fig. 5a). Compared with N₂

saturated solution, a significantly higher current density was observed in the CO₂ saturated solution, indicating the superior CO₂ reduction activity of COFs (Supplementary Fig. 28). The LSV curves showed that the current densities of the CoTAPP-PATA-COF and NH-COF were close in the same potential range. In comparison, the current densities of N⁺-COF and N⁺-NH-COF were increased, suggesting that the ionized skeletons promoted electron transfer and enhanced the current density (Fig. 5a). The corresponding Tafel slope of the CoTAPP-PATA-COF was 236 mV dec⁻¹, which declined to 203, 197, and 184 mV dec⁻¹ for N⁺-COF, NH-COF, and N⁺-NH-COF, respectively (Fig. 5b). It suggests that the post modifications significantly improve the electrocatalytic CO₂RR kinetics[33].

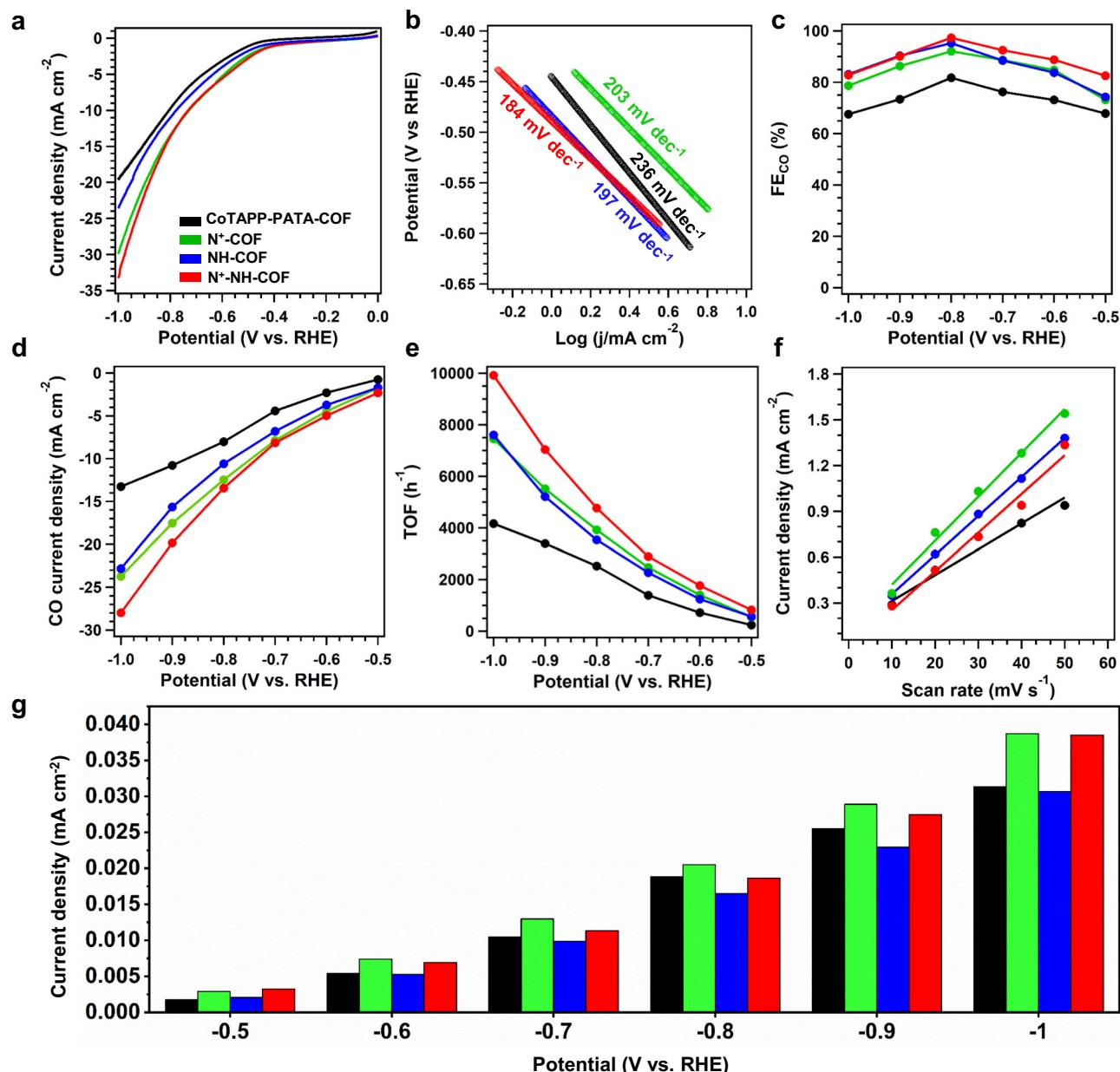

**Fig. 5 | Electrocatalysis CO₂ reduction reaction on COFs. a** LSV curves, **b** Tafel slopes, **c** CO faradaic efficiency, **d** the partial CO current density, **e** the corresponding TOF values, **f** the ECSA slopes and **g** the CO current density by the normalized ECSA for CoTAPP-PATA-COF (black), N⁺-COF (green), NH-COF (blue) and N⁺-NH-COF (red) from −0.5 to −1.0 V in 0.5 M KHCO₃ under CO₂ atmosphere.

The reduction products of CO₂RR were analysed via the gas and liquid chromatography, and only CO and H₂ were detected. The constant time-dependent total geometric current densities of the CoTAPP-PATA-COF, N⁺-COF, NH-COF, and N⁺-NH-COF were determined for each potential between −0.5 and −1.0 V for 1000 s, indicating that the COF catalysts had good stability (Supplementary Figs. 29–32). The CO Faradaic efficiencies (FE$_{CO}$) of the CoTAPP-PATA-COF were 67.84%, 73.11%, 76.26%, 81.75%, 73.42%, and 67.54% at −0.5, −0.6, −0.7, −0.8, −0.9, and −1.0 V, respectively (Fig. 5c, black curve). In addition to the selectivity, the activity was revealed by the partial CO current density (j$_{CO}$) of the catalyst. The CoTAPP-PATA-COF exhibited the highest j$_{CO}$ of 13.26 mA cm⁻² at a potential of −1.0 V (Fig. 5c, black curve). The CO selectivity was remarkably improved on the COFs that contain C−N linkages and /or skeleton ionization. The N⁺-COF had considerably higher selectivity than the base COF at the same potential, delivering FE$_{CO}$ of 73.12%, 84.75%, 88.71%, 92.07%, 86.31%, and 78.66% at −0.5,

−0.6, −0.7, −0.8, −0.9, and −1.0 V, respectively (Fig. 5c, green curve). The highest j$_{CO}$ on N⁺-COF was 23.73 mA cm⁻² at −1.0 V, which was higher than that of the base COF (Fig. 5d, green curve). The FE$_{CO}$ for the NH-COF were 74.26%, 83.75%, 88.49%, 95.26%, 90.31%, and 83.12%, which are higher than that of CoTAPP-PATA-COF and N⁺-COF at the same potentials (Fig. 5c, blue curve). The highest j$_{CO}$ on NH-COF was 22.83 mA cm⁻² at −1.0 V, being also higher than that of base COF (Fig. 5d, blue curve). In comparison, the N⁺-NH-COF achieved FE$_{CO}$ of 82.56%, 88.76%, 92.51%, 97.32%, 90.12%, and 82.78% from −0.5 to −1.0 V (Fig. 5c, red curve) and the highest j$_{CO}$ of 28.01 mA cm⁻² at −1.0 V (Fig. 5d, red curve), which are higher than those of other COFs. Therefore, the C−N linkages improved the catalytic selectivity, and the ionic skeleton contributed to higher activity. The CO production turnover frequencies (TOFs) of the prepared COFs were obtained at different potentials by accounting the amount of CoTAPP units as electrocatalytically active sites. The highest TOF values obtained for

the CoTAPP-PATA-COF, N⁺-COF, NH-COF, and N⁺-NH-COF were 4166.19, 7453.19, 7604.63, and 9922.68 h⁻¹ at −1.0 V, respectively (Fig. 5e). As the N⁺-NH-COF yielded the highest TOF in compared to other COFs, it demonstrated that the multilevel post modification boosted both $CO_2RR$ activity and CO selectivity of the base COF.

The electrochemical double layer capacitances ($C_{dl}$) were calculated using the cyclic voltammogram (CV) plots in a potential range of −0.16 to −0.36 V at the scan rates of 10−50 mV s⁻¹ (Supplementary Fig. 33). Correspondingly, the $C_{dl}$ values for the CoTAPP-PATA-COF, N⁺-COF, NH-COF, and N⁺-NH-COF were 16.92, 26.34, 24.37, and 28.76 mF cm⁻², respectively (Fig. 5f)[38]. To investigate the exposed active sites, the electrochemically active surface areas (ECSAs) of the COFs were then calculated by ECSA = $C_{dl}/C_s$, where the $C_s$ is 0.04 mF cm⁻² [65]. N⁺-NH-COF offered the highest ECSA (719) relative to CoTAPP-PATA-COF (423), N⁺-COF (659), and NH-COF (609), which is in line with its $CO_2RR$ performance. Moreover, the ECSA of N⁺-COF was higher than the base COF and NH-COF, suggesting that the ionic frameworks provided more active sites[65]. The normalized current densities on the COFs, i.e., $j_{CO}$ per ECSA, were further evaluated (Fig. 5g). Within the potential range of −0.5 to −1.0 V, the ionized COFs of N⁺-COF and N⁺-NH-COF always exhibited the higher normalized current density than that of un-ionized COFs of CoTAPP-PATA-COF and NH-COF, confirming that the ionization helps to improve the COF activity towards $CO_2RR$. Furthermore, the Nyquist plots showed that the charge transfer resistances ($R_{ct}$) over N⁺-COF and N⁺-NH-COF were 36 and 24 Ω, respectively, being smaller than that of CoTAPP-PATA-COF (52 Ω) and NH-COF (42 Ω). This suggested that the ionization of skeletons enhanced the charge transfer capacity of COFs (Supplementary Fig. 34).

Moreover, the long-term stability of the N⁺-NH-COF was evaluated at −0.8 V in $CO_2$-saturated KHCO₃ for 25 h (Supplementary Fig. 35). The $FE_{CO}$ kept stable around 97% and the relative current density ($j/j_0$) was >94.5% within 25 h, indicating the excellent long-term stability of N⁺-NH-COF for $CO_2RR$. To confirm the stability, the used N⁺-NH-COF was checked by the PXRD, FTIR and XPS measurements after the long-term stability test. The PXRD patterns showed that all peaks were fully retained, and peaks were well-retained after the long-term stability test (Supplementary Fig. 36). The FTIR spectra showed that all the peaks of the N⁺-NH-COF were well maintained (Supplementary Fig. 37). In addition, the Co 2p, I 3d and N 1s spectra showed the same states to that of fresh sample (Supplementary Figs. 38–40). Thus, the structure of the N⁺-NH-COF was retained.

### Density functional theory calculations

To further understand the different performance of CoTAPP-PATA-COF, N⁺-COF, NH-COF and N⁺-NH-COF for $CO_2RR$, the density functional theory (DFT) calculations were carried out at the theoretical level of CAM-B3LYP/6-311 G(d) (SDD for Co) for the cluster models of these four COFs. Reasonable geometrical, electronical and thermodynamic information could be obtained with cluster models for COFs and similar systems[66–68]. All structures along the potential energy surfaces were optimized without any restrictions. Frequencies were further performed to confirm that all optimized geometries are local minima and to obtain the Gibbs free energies. In the calculation, the four model molecules were labelled as M-CoTAPP-PATA-COF, M-I-N⁺-COF, M-NH-COF and M-I-N⁺-NH-COF, respectively, where the counter ion I⁻ was considered for the N⁺-COF and N⁺-NH-COF (Fig. 6a).

As shown in Fig. 6b and Supplementary Fig. 41, $CO_2RR$ includes four steps: the absorption of $CO_2$, formation of *COOH and *CO where a proton coupled a single electron transfer take place for each step, and CO desorption from the active metal centre. From $CO_2$ to the product CO, the $CO_2RR$ catalysed by M-I-N⁺-COF and M-I-N⁺-NH-COF are slightly endergonic, while those catalysed by M-CoTAPP-PATA-COF and M-NH-COF are exergonic. This is in accordance with the excellent catalytic ability of the four COFs. The relative free energies of *COOH

formation are higher than the initial state of $CO_2$ (i.e., the relative zero point of free energies), while the process from *COOH to the final product CO are largely exergonic. This indicates that the formation of *COOH is the rate control step. The lowest free energy changes of *COOH state for M-I-N⁺-NH-COF is in line with its best performance from a thermodynamic viewpoint. In addition, both M-I-N⁺-COF and M-I-N⁺-NH-COF have the lower free energy change values (ΔG) of *COOH than those of M-CoTAPP-PATA-COF and M-NH-COF, which indicates a stronger promotion effect of ionization than the neutral ones. As shown in Fig. 6c–e, we provided the main geometrical parameters and the Mulliken charges on main atoms for stationary points along the reaction pathway of M-I-N⁺-NH-COF catalysing $CO_2RR$. The oxygen $O_1$ atom in $CO_2$ coordinates with Co atom in M-I-N⁺-NH-COF. The Co-O1 bond is 2.734 Å and the charge transfer from M-I-N⁺-NH-COF to $CO_2$ is negligible because the total charge on $CO_2$ is nearly zero (Fig. 6c). The formation of *COOH by the proton transfer makes the distortion of $CO_2$ portion with the angle of $O_1$-C-$O_2$ decreases from 178.6° to 121.7°, along with the elongation of C-$O_2$ bond to from 1.154 Å to 1.343 Å (Fig. 6d). The shortened Co-C bond results a stronger charge transfer between M-I-N⁺-NH-COF and $CO_2$. The departure of *OH leads to the formation of *CO, where Co is of a vertical orientation with a Co-C-O bond angle of 168.1° (Fig. 6e). In addition, the main geometrical parameters and the Mulliken charges on main atoms for stationary points along the reaction pathway of CoTAPP-PATA-COF catalysing $CO_2RR$ was shown in Supplementary Fig. 42. As the same as M-I-N⁺-NH-COF, $CO_2$ was adsorbed with the coordination between Co and oxygen, while the intermediates of *COOH and *CO were coordinated with Co with the C atoms in the case of CoTAPP-PATA-COF calculations.

Furthermore, in Fig. 6f–i, we listed the partitional density of states (PDOSs) of Co (blue lines), COOH (red lines) and remainder of *COOH (green lines) for adsorption state of COOH on M-CoTAPP-PATA-COF, M-I-N⁺-COF, M-NH-COF and M-I-N⁺-NH-COF. It could be found that the contribution of Co is larger in M-I-N⁺-COF and M-I-N⁺-NH-COF than in M-CoTAPP-PATA-COF and M-NH-COF, indicating that the introducing of methyl groups strengthens the electronic density on Co atom. This may promote the interaction between the Co and COOH portion during the reaction. All above calculated results are in consistent with the experimental observations.

## Discussion

In this study, a multilevel post-function strategy was demonstrated to modulate the properties of COFs (porosity, crystallinity, and electron states), which can contribute to their tuneable catalytic performance for $CO_2RR$. By constructing catalytic COFs with ionic and NH linkers, the catalytic COFs allow catalysing $CO_2RR$ with high activity and a maximum TOF value of 9922.68 h⁻¹ at −1.0 V, and high selectivity with the highest $FE_{CO}$ of 97.32% at −0.8 V. This work provides us a more in-depth understanding of COFs and their applications in electrochemical energy storage and conversion systems. Meanwhile, it also guides us to construct multilevel post-synthetic modification COFs for achieving both tailored activity and high stability.

## Methods

### Synthesis of CoTAPP-PATA-COF

A mixture of 1-butanol / 1,2-dichlorobenzene (0.5 mL/0.5 mL), (PATA (14.2 mg, 0.03 mmol), TAPP(Co) (23.5 mg, 0.03 mmol), and an aqueous acetic acid solution (6 M, 0.1 mL) was degassed in a Pyrex tube (10 mL) by three freeze-pump-thaw cycles. The tube was sealed and heated at 120 °C for 3 days. The precipitate was collected by centrifugation, washed with tetrahydrofuran, and dried at 120 °C under vacuum overnight to give CoTAPP-PATA-COF in a yield of 87.6%.

### Synthesis of NH-COF

A mixture of CoTAPP-PATA-COF (50 mg), NaBH₄ (100 mg), and methanol (50 mL) was stirred in a 100 mL flask for 12 h at 0 °C. The

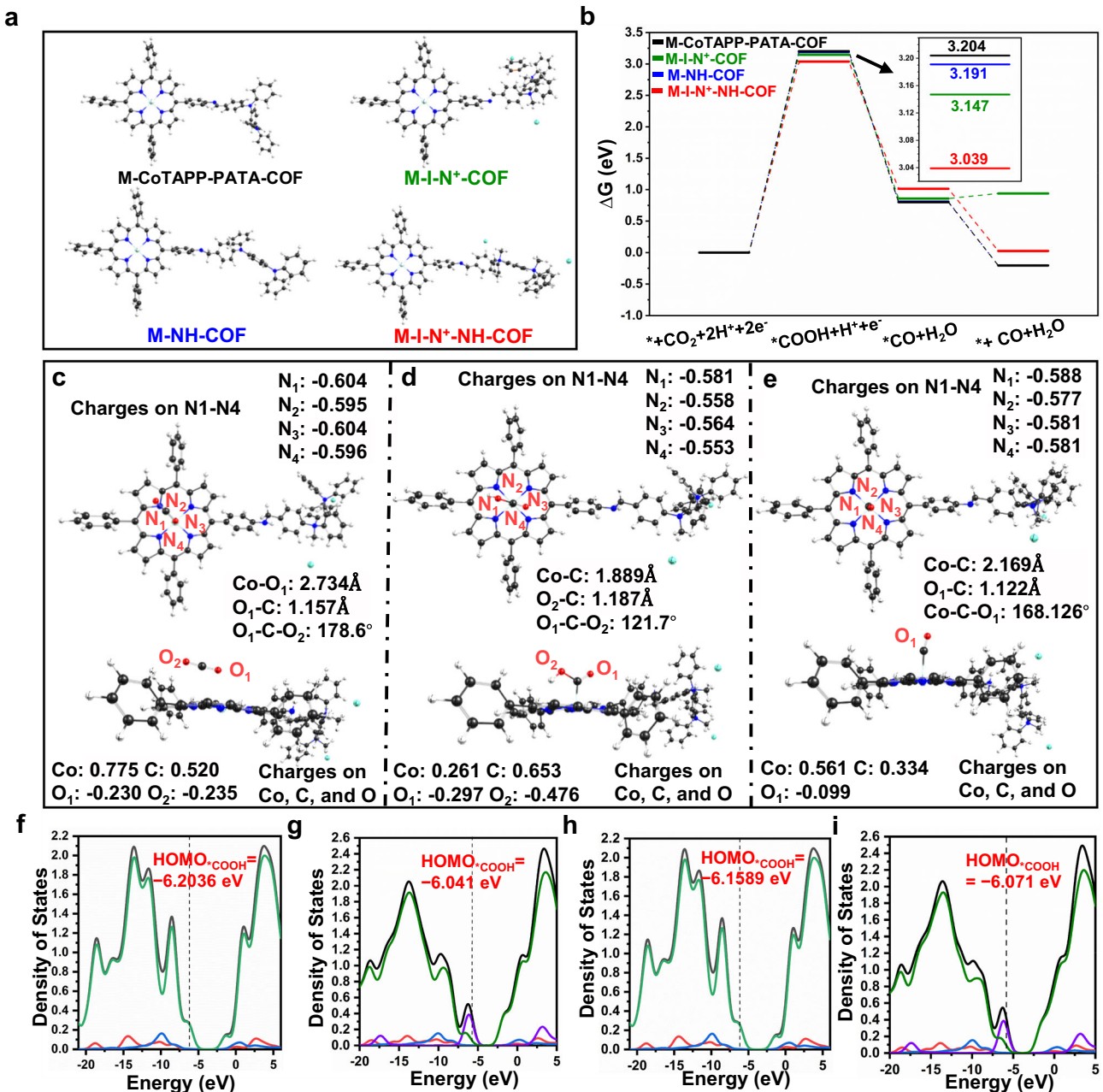

**Fig. 6 | Theorical calculation. a** The optimized geometrical structures and **b** Calculated free energy change (ΔG) diagram of M-CoTAPP-PATA-COF, M-I-N⁺-COF, M-NH-COF and M-I-N⁺-NH-COF catalysing CO₂RR (M-Model). The optimized geometrical structures of stationary points along the reaction pathway of M-I-N⁺-NH-COF catalyzing CO₂RR reaction, **c** CO₂*, **d** COOH*, **e** CO*, along with the main geometrical parameters and the Mulliken charges on main atoms. The black, red, white, blue, light cyan and dark cyan balls represent carbon, oxygen, hydrogen, nitrogen, cobalt and iodine atoms, respectively. The total density of states (TDOSs, black lines), partitional density of states (PDOSs) of Co (blue lines), COOH (red lines), remainder of COOH* (green lines) and counter ion I⁻ (purple line) for **f** M-CoTAPP-PATA-COF, **g** M-I-N⁺-COF, **h** M-NH-COF and **i** M-I-N⁺-NH-COF catalysing CO₂RR.

mixture was then poured into a large amount of ice water. After filtration, the white solid was washed three times with water and dried.

### Synthesis of N⁺-COF

The cool stored 5 mL iodomethane (CH₃I) liquid was continuously added into a solution of the CoTAPP-PATA-COF (50 mg) in ethanol (200 mL) to be magnetically stirred at ambient temperature. Afterward, the mixture was vibrated at 60 °C for 24 h. The obtained white precipitate was collected and washed with ether followed by drying in vacuum for 6 h.

### Synthesis of N⁺-NH-COF

A mixture of N⁺-COF (50 mg), NaBH₄ (100 mg, 2.64 mmol), and methanol (50 mL) was stirred in a 100 mL flask for 12 h at 0 °C. The mixture was then poured into a large amount of ice water. After filtration, the white solid was washed three times with water and dried.

## Data availability

All data supporting the findings of this study are available within the article, as well as the Supplementary Information file. All other data supporting the findings of the study are available from the corresponding author upon request.

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

## Acknowledgements

The authors acknowledge the financial support from the Natural Science Foundation of Shanghai (20ZR1464000 (Q.X.) and 22ZR1470100 (G.Z.)), the National Natural Science Foundation of China (21878322 (G.Z.), 22075309 (G.Z.)), the Youth Innovation Promotion Association of Chinese Academy of Sciences (Q.X.), and the Biomaterials and Regenerative Medicine Institute Cooperative Research Project (2022LHA09 (G.Z.)), Shanghai Jiaotong University School of Medicine. The Funding for Internationalization Training HighLevel Talent in Henan Province and Open Project of ZhengZhou JiShu Institute of AI Science (ZZJSA2023001 (C.-X.C.)).

## Author contributions

Q.X. conceived the idea and designed the experiments. M.L. and S.Y. performed the experiments. C.-X.C. contributed to the theory calculation part. X.Y., G.L. and X.L. participated in some experiments. M.L., J.H., G.-Z.C., Q.X. and G.Z. wrote and revised the manuscript. All the authors contributed to the data interpretation, discussion, and manuscript revision. All authors have given approval to the final version of the manuscript. M.L. and S.Y. contributed equally to this work.

## Competing interests

The authors declare no competing interests.

## Additional information

[1]CAS Key Laboratory of Low-Carbon Conversion Science and Engineering, Shanghai Advanced Research Institute, Chinese Academy of Sciences, Shanghai
201210, P. R. China. [2]Department of Chemical and Environmental Engineering, University of Nottingham Ningbo China, Ningbo 315199, P. R. China. [3]School of
Physical Science and Technology, ShanghaiTech University, Shanghai 201210, P. R. China. [4]School of Chemical Engineering, University of Chinese Academy
of Sciences, Beijing 100049, P. R. China. [5]School of Chemistry and Chemical Engineering, Institute of Computational Chemistry, Henan Institute of Science
and Technology, Xinxiang 453003, P. R. China. [6]ZhengZhou JiShu Institute of AI Science, Zhengzhou 451162, P. R. China. [7]Department of Chemical and
Environmental Engineering, University of Nottingham, Nottingham NG7 2RD, UK. [8]These authors contributed equally: Minghao Liu, Shuai Yang.
✉e-mail: chengxingcui@hist.edu.cn; xuqing@sari.ac.cn; zenggf@sari.ac.cn

