## [Peer Review File · Nature Communications]

Post-synthetic modification of covalent organic frameworks for CO₂ electroreductionREVIEWER COMMENTS

Reviewer #1 (Remarks to the Author):

Authors have done post-synthetic modification in covalent-organic framework for electrochemical CO₂ reduction. Research on COF for the utilization of electrochemical CO₂ reduction is well explored. Upon modifying the inactive sites of this COF, the authors obtained N⁺-NH-COF covalent organic frameworks, which demonstrated a decent degree of CO selectivity in CO₂ reduction reactions, with a maximum Faradaic efficiency (FECO) of 97.32%. This work has been carried out thoroughly. The scientific nature of this work meets the requirements of the Nature Communication. Considering the novelty of this work, I would like to recommend its publication. Here are some questions should be solved to improve the quality of this research:

1. Characterization methods such as XPS should be used to further investigate the anionic species in the N⁺-NH-COF catalyst that neutralize the positive charge, and determine whether these are I⁻ ions. It would also be significant to investigate whether the anionic composition of the catalyst changes before and after the catalytic reaction, and provide experimental evidence to support this.
2. To compare the charge transfer resistance (R_{ct}) of CoTAPP-PATA-COF, N⁺-COF (green), NH-COF and N⁺-NH-COF samples, the Nyquist plots of all samples should be fitted by a typical equivalent circuit.
3. The thermal stability of catalysts is an important characteristic, please provide the TGA analysis for these COFs.
4. The authors should provide other characterizations or absorption test for the hydrophobicity of N⁺-NH-COF.
5. There are some grammar mistakes in the manuscript. Please carefully check the scientific writing.
6. There are some important works about cationic COFs and MOFs for electrocatalysis application, such as *Angew. Chem. Int. Ed.* 2022, e202215687; *National Science Review* 9: nwab157, 2022.

Reviewer #2 (Remarks to the Author):

The manuscript contributed by Liu et al. developed a two-step post-synthetic approach for covalent organic frameworks (COFs). The obtained COFs were employed as catalysts for electrocatalytic CO₂ reduction reaction (CO₂RR). However, the post-synthetic approaches have been reported previously by other groups (*Chem* 2018, 4, 1696-1709; *J. Am. Chem. Soc.* 2019, 141, 14433–14442). The authors just combined above-mentioned post-synthetic together in one reported COFs (*Angew. Chem. Int. Ed.* 2022, 407, e202213522). Obviously, the so-called “Multilevel Post-synthetic Modification” strategy is lack of new idea and creativity. In addition, the obtained porphyrin-based N⁺-NH-COF exhibited a CO Faradaic efficiency (FECO) of 97.32% and a turnover frequency (TOF) of 9922.68 h⁻¹. The porphyrin-based CO₂RR catalysts have been widely studied (*Chem. Soc. Rev.* 2023, DOI: 10.1039/D2CS00843B). Most of these materials exhibit quite close performance, for example, with FECO values from 85%-97%. The CO₂RR performance of N⁺-NH-COF is very similar with that of many other porphyrin- or phthalocyanine-based COFs (e.g. *Small*, 2021, 17, 2004933; *Angew. Chem., Int. Ed.*, 2020, 59, 16587-16593; *Angew. Chem., Int.*

Ed., 2021, 60, 4864-4871). Overall, the manuscript is poorly organized and is not suitable for the wide readership of Nature Communications.

Reviewer #3 (Remarks to the Author):

In this manuscript, the authors demonstrate the utilization of the double modifications of the CoTAPP-PATA-COF for the regulation of selectivity and activity of the electrocatalytic CO₂ reduction reactions. The linkages and linker have been respectively modified by reduction and ionization reactions. The authors have used FTIR and XRD to characterize the synthesis of the pristine COF and the COFs after post-modifications. The results showed that the post-synthetic modification strategy can adjust the performance of CO₂RR. Notably, the dual-functional COF achieved high selectivity with a maximum CO Faradaic efficiency of 97.32% and high turnover frequencies of 9922.68 h⁻¹. Despite this, the manuscript is still short of a few fundamental aspects regarding the structural characterizations and interpretation of the mechanism for the observation of modified performance. The reviewer suggests that this manuscript is too premature to be published before the following issues get properly addressed.

1. The authors have given a few assumptions in the whole manuscript. These assumptions are not exactly consistent which makes it hard for the reader to understand which is essentially the underlying mechanism for the improved performance. For example, in the abstract, the authors demonstrate “Amine-linkages favour molecular diffusion, and the ionization skeletons enable to delocalize charges to construct a protected ion transport pathway”. In lines 130-132, the authors indicate that the ionization for the skeleton can disturb the interlayer stacking because of the electrostatic repulsion and steric hindrance, and promote the exposure of active sites and provide more space for the electrochemical reaction. In line 197, the authors suggested ionization units could improve the activity and the C-N bonds could promote CO₂ adsorption. The authors need to carefully articulate their key hypothesis.

2. Further evidence is needed for the implementation of the dual-level modification.

- a) The authors indicated in Figure S1 that -C=N was weakened by reduction, and it was unreasonable to attribute the remaining peak to -C=N in CoTAPP. The peak of pyrrole -C=N FTIR in porphyrins was usually 1340cm⁻¹, and the author needed to indicate the position of -C-N in the IR spectrum. Solid-state ¹³C NMR could be a good technique to further support whether -C=N is successfully reduced to -C-N.
- b) Given that the methylation was done with a large excess of CH₃I, the readers would wonder why only the N atoms in PATAN were methylated instead of N atoms in imine bonds.

3. The authors proved the adsorption and binding of CO₂ by post-modification, but the actual CO₂ adsorption value measured is contrary to the author's statement. The author defined that the adsorption of CO₂ by different COF is unreasonable by $n = \text{SCO}_2/\text{SBET}$. This is because the performance of the electrocatalyst CO₂RR is tested using catalysts with the same mass rather than the same specific surface areas.

4. XPS analysis:

- a) The signal-to-noise ratio of XPS data (Figure 3a) for Co 2p is too low to do any convincing interpretation. The fitting of Co 2p is very poor as well.
- b) The author stated that Co 2p showed about 0.13 eV blue shift due to the electron-withdrawing effect, however, it was exactly the opposite of the data as shown in Figure 3a. It seems that there is a redshift after N is methylated.

5. The conductivity of the material can not solely determined by the band gap, and it needs to be independently determined by conductivity measurement.

6. The electrocatalytic performance presented by the author in the manuscript diagram is inconsistent with the performance described in the manuscript, which needs to be further confirmed.

- a) The authors suggest that -C-N linkages improved the catalytic activity and the ionic linker contributed to higher selectivity, contrary to the results in Figure 4c, where -C-N linkages appear to increase selectivity and ionic linkers appear to increase activity.
- b). ECSA data show that NH-COF > N+-NH-COF, which is opposite to the reactivity shown in Figure 4d, please confirm.

7. Theoretical calculations need to be further confirmed.

- a) The authors omitted many important details of the DFT calculation part, which can affect its reliability. The authors should clearly mention that the calculations were done based on the molecular models, which are hydrogen-terminated fragments of the COFs, instead of using the COFs themselves. Correspondingly, the naming of the materials in Figure 5a should be revised. The authors should explain in the manuscript why the use of such molecular models is reasonable and point out how this simplification may affect the results. Since N+-COF and N+-NH-COF are positively charged, in the calculations, how are the corresponding counterions of the positively charged species treated? It is also not known if the authors have incorporated the thermodynamic corrections in their calculation.
- b) The reviewer noticed that in the energy profile (Figure 5a), the free energy differences for the same kind of intermediate from different catalysts are very small. For example, the energy difference of COOH* mediated by the four types of catalysts is smaller than 0.04 eV. Such small energy differences, which may be within the error of the theoretical calculations, cannot be used as support to correlated with the experimental results of the catalysts.
- c) Is the calculation of free energy in Figure 5A in the manuscript reasonable? i) The whole energy profile is going down which makes the reader wonder that CO₂RR is a spontaneous reaction; ii) For Figure 5A, it seems the formation of the CO* is the rate-determining step? iii) The free energy change from CO₂ to CO* is too large than normal. The reviewer suggests that the authors carefully check their calculation method.

8. The whole manuscript needs to be checked carefully. There are many mismatches between the material name and the corresponding data in the article, which will cause misunderstandings among the readers. For example, supplementary Figure S25 and Supplementary Figure S27 are both for N+-COF. One of them must be incorrectly labeled. The red line in the Figure S30, which is for FEco is in the exact

same shape as the black line. The FE data is wrongly plotted.

9. Other issues:

a) TAPP and PATA, as simplified names, need to be defined the first time they appear in the manuscript.

b) Line 127-128: "Their highly crystal structures...." should be "Their highly crystalline structures...".

c) In lines 228-229, "the NH-COF showed the highest Tafel slopes than that of other COFs, indicating the superior combination with CO₂ in the process of CO₂RR" reads confusing. The highest Tafel slopes indicate that the kinetics are most unfavorable.

d) In Supplementary Table S5, the standard for the comparison of performance parameters is not uniform. For example, the author gives FE (97.37%) when the potential is -0.8 V, but J_{co} (21.37) is given when the potential -0.9 V for the same material (N+-NH-COF).

e) In the abstract "OOH*" intermediates should be "COOH*".

Reply to Referees' Comments (NCOMMS-23-02659)

We truly appreciate the reviewers for their highly professional and thoughtful comments, which offer us a precious opportunity to enhance this work and deepen its scope. We have carefully revised the manuscript considering these suggestions. All the changes to the manuscript as well as the Supplementary Materials were marked in colour in the revision. In this Response to Comments file, the original comments of reviewers are copied in black while our responses are provided to the comments one-to-one and marked in blue.

Reviewer #1:

Authors have done post-synthetic modification in covalent-organic framework for electrochemical CO₂ reduction. Research on COF for the utilization of electrochemical CO₂ reduction is well explored. Upon modifying the inactive sites of this COF, the authors obtained N⁺-NH-COF covalent organic frameworks, which demonstrated a decent degree of CO selectivity in CO₂ reduction reactions, with a maximum Faradaic efficiency (FECO) of 97.32%. This work has been carried out thoroughly. The scientific nature of this work meets the requirements of the Nature Communication. Considering the novelty of this work, I would like to recommend its publication.

=====

Response:

We're thankful for the strong supports and the constructive comments provided from Reviewer #1. We have carefully revised the manuscript considering the thoughtful suggestions of the reviewer. The details of our responses to the comments were listed below.

=====

Here are some questions should be solved to improve the quality of this research:

1. Characterization methods such as XPS should be used to further investigate the anionic species in the N(+)-NH-COF catalyst that neutralize the positive charge, and determine whether these are I-ions. It would also be significant to investigate whether the anionic composition of the catalyst changes before and after the catalytic reaction, and provide experimental evidence to support this.

=====

Response:

We appreciate this comment.

In the revision, the N⁺-NH-COF samples before and after the CO₂RR test were measured by XPS. As shown in Supplementary Fig. 40, the XPS I 3d spectrum of the fresh N⁺-NH-COF indicates I

$3d_{5/2}$ peak at 630.5 eV and $3d_{3/2}$ at 619.0 eV, assigned to iodide ions state (*Angew. Chem. Int. Ed.* **2023**, *135*, e202214449). For the used N^+ -NH-COF sample, no visible changes are observed, suggesting that I^- state is maintained after CO_2RR .

The above results were provided in the Supplementary Materials and discussed in the Main Text.

Supplementary Figure 40. The XPS I 3d spectra of N^+ -NH-COF before and after the CO_2RR .

2. To compare the charge transfer resistance (R_{ct}) of CoTAPP-PATA-COF, $N(+)$ -COF (green), NH-COF and $N(+)$ -NH-COF samples, the Nyquist plots of all samples should be fitted by a typical equivalent circuit.

Response:

We appreciate this comment.

Supplementary Figure 34. EIS spectra of CoTAPP-PATA-COF (black), N^+ -COF (green), NH-COF (blue) and N^+ -NH-COF (red) (insert: typical equivalent circuit).

In the revision, the Nyquist plots were fitted by a typical equivalent circuit.

As shown in Supplementary Figure 34, the charge transfer resistances (R_{ct}) over N^+ -COF and N^+ -NH-COF were 36 and 24 Ω , respectively, being smaller than that of CoTAPP-PATA-COF (52 Ω) and NH-COF (42 Ω). This suggested that the ionization of skeletons enhanced the charge transfer capacity of COFs.

The above results were provided in the Supplementary Materials and discussed in the Main Text.

3. This is an important characteristic, please provide the TGA analysis for these COFs.

Response:

We appreciate this comment.

In the revision, the thermal stability of these COFs was measured by the thermal gravimetric analysis (TGA) from 100 to 800 $^{\circ}C$ in N_2 . As shown in Supplementary Fig. 22, the post-modified COFs kept the similar thermal stability to that of base COF, and no significant mass loss was observed at the temperature < 480 $^{\circ}C$ in N_2 . This reveals that the thermal stability of COFs was well maintained after the post-modifications.

Supplementary Figure 22. TGA curves of CoTAPP-PATA-COF (black), N^+ -COF (green), NH-COF (blue) and N^+ -NH-COF (red) under N_2 atmosphere.

The above results were provided in the Supplementary Materials and discussed in the Main Text.

4. The authors should provide other characterizations or absorption test for the hydrophobicity of

N(+)-NH-COF.

Response:

We appreciate this comment.

In the revision, the hydrophobic behavior of N⁺-NH-COF was further verified by water uptake test.

As shown in Supplementary Fig. 24, the N⁺-NH-COF exhibited a low water vapour uptake capacity of 188 cm³ g⁻¹ at 298 K and an inflection point at high relative pressures ($P/P_0 = 0.5$), which are in good agreement with the hydrophobic behaviors (*Adv. Mater.* **2018**, *30*, 1704304). For the electrocatalysis of CO₂RR, the hydrophobic nature of catalyst is crucial to protect the active sites through suppressing the competitive adsorption of water, leading to the improvements of selectivity and energy efficiency (*Adv. Mater.* **2022**, *34*, 2205186).

Supplementary Figure 24. The water uptakes of N⁺-NH-COF at 298 K.

The above results were provided in the Supplementary Materials and discussed in the Main Text.

5. There are some grammar mistakes in the manuscript. Please carefully check the scientific writing.

Response:

We appreciate this comment.

The manuscript has been revised carefully and thoroughly.

6. There are some important works about cationic COFs and MOFs for electrocatalysis application,

such as *Angew. Chem. Int. Ed.* 2022, e202215687; *National Science Review* 9: nwab157, 2022.

Response:

We appreciate this comment.

The above-mentioned references were cited in the revision. In addition, more important references were also cited, which were listed as followings:

22. He, C. et al. Metal-organic frameworks bonded with metal N-heterocyclic carbenes for efficient catalysis. *Natl. Sci. Rev.* **9**, nwab157 (2022).

23. Large-area Free-standing Metalloporphyrin-based Covalent Organic Framework Films by Liquid-air Interfacial Polymerization for Oxygen Electrocatalysis. *Angew. Chem. Int. Ed.* **135**, e202214449 (2023).

48. El-Kaderi, H. M. et al. Designed Synthesis of 3D Covalent Organic Frameworks. *Science* **316**, 268-272 (2007).

49. Uribe-Romo, F. J. et al. A Crystalline Imine-Linked 3-D Porous Covalent Organic Framework. *J. Am. Chem. Soc.* **131**, 4570-4571 (2009).

50. Fang, Q. et al. 3D Microporous Base-Functionalized Covalent Organic Frameworks for Size-Selective Catalysis. *Angew. Chem. Int. Ed.* **53**, 2878-2882 (2014).

61. Liu, H. et al. Covalent Organic Frameworks Linked by Amine Bonding for Concerted Electrochemical Reduction of CO₂. *Chem* **4**, 1696-1709 (2018).

62. Mi, Z. et al. Stable Radical Cation-Containing Covalent Organic Frameworks Exhibiting Remarkable Structure-Enhanced Photothermal Conversion. *J. Am. Chem. Soc.* **141**, 14433-14442 (2019).

63. Kalmutzki, M. J., Diercks, C. S. & Yaghi, O. M. Metal–Organic Frameworks for Water Harvesting from Air. *Adv. Mater.* **30**, 1704304 (2018).

64. Zhang, X. et al. Insight Into Heterogeneous Electrocatalyst Design Understanding for the Reduction of Carbon Dioxide. *Adv. Energy Mater.* **12**, 2201461 (2022).

65. Li, N. et al. Boosting Electrocatalytic CO₂ Reduction with Conjugated Bimetallic Co/Zn Polyphthalocyanine Frameworks. *CCS Chem.* **5**, 1130-1143 (2022).

66. Mandal, M., Cramer, C. J., Truhlar, D. G., Sauer, J. & Gagliardi, L. Structure and Reactivity of Single-Site Vanadium Catalysts Supported on Metal–Organic Frameworks. *ACS Catal.* **10**, 10051-10059, (2020).
67. Zhai, L. et al. CoN₅ Sites Constructed by Anchoring Co Porphyrins on Vinylene-Linked Covalent Organic Frameworks for Electroreduction of Carbon Dioxide. *Small* **18**, 2200736, (2022).
68. Lin, C. et al. Covalent Organic Frameworks with Tailored Functionalities for Modulating Surface Potentials in Triboelectric Nanogenerators. *Angew. Chem. Int. Ed.* **61**, e202211601, (2022).

Reviewer #2:

The manuscript contributed by Liu et al. developed a two-step post-synthetic approach for covalent organic frameworks (COFs). The obtained COFs were employed as catalysts for electrocatalytic CO₂ reduction reaction (CO₂RR). However, the post-synthetic approaches have been reported previously by other groups (Chem 2018, 4, 1696-1709; J. Am. Chem. Soc. 2019, 141, 14433–14442). The authors just combined above-mentioned post-synthetic together in one reported COFs (Angew. Chem. Int. Ed. 2022, 407, e202213522). Obviously, the so-called “Multilevel Post-synthetic Modification” strategy is lack of new idea and creativity. In addition, the obtained porphyrin-based N⁺-NH-COF exhibited a CO Faradaic efficiency (FECO) of 97.32% and a turnover frequency (TOF) of 9922.68 h⁻¹. The porphyrin-based CO₂RR catalysts have been widely studied (Chem. Soc. Rev. 2023, DOI: 10.1039/D2CS00843B). Most of these materials exhibit quite close performance, for example, with FECO values from 85%-97%. The CO₂RR performance of N⁺-NH-COF is very similar with that of many other porphyrin- or phthalocyanine-based COFs (e.g. Small, 2021, 17, 2004933; Angew. Chem., Int. Ed., 2020, 59, 16587-16593; Angew. Chem., Int. Ed., 2021, 60, 4864-4871). Overall, the manuscript is poorly organized and is not suitable for the wide readership of Nature Communications.

=====

Response:

We're very thankful for the highly valuable and insightful questions from Reviewer #2, which have certainly helped us to improve the quality of this manuscript. The manuscript was carefully revised based on these suggestions. Additional experiments and reasonable analysis were performed to provide solid supports in the revision.

✓ **Innovation of this work and differences to refs.**

To response the comment raised above, we explained the innovation of this work and the difference of our work than the above-mentioned references from the following aspects:

First of all, no two-step / multilevel post-modification of COFs was reported so far. This work is the first example to endow the original COF material with multifunction by the multilevel post-modification.

Significance of the innovation on multilevel post-modification was demonstrated in this work. We proved that the multilevel post-modification is highly effective to change the reactivity of COFs for a certain reaction. The base COF, CoTAPP-PATA-COF, is previously reported as a bifunctional ORR / OER catalyst but with a poor CO₂RR reactivity (FECO 81%, TOF 4266 h⁻¹, j_{CO} 15 mA cm⁻²), which is successfully changed to a highly reactive CO₂RR catalyst (FECO 97%, TOF 9923 h⁻¹, j_{CO}

21 mA cm⁻²) by the multilevel post-modification. Therefore, our work provided a practicable methodology to extend the functions and applications of the existed COFs. With this strategy, it is also potential to develop new / special functions on the numerous existed COFs.

Various structure and composition are the advantages of COFs over other porous crystalline materials, resulting in developing multitudinous COFs in the past decade. Unlike SAPO-34, for example, is well known as a MTO catalyst and also a CO₂ separation material in the zeolite family, or ZIF-8 is a classic material for paraffin/ olefin separation in the area of MOFs, the variability of COFs in structure and composition limits the concentration of research community to some extent to develop classic and representative COFs. In contrast, our work tries to endow the existed COF with new application, which is helpful to screen the classic and representative COFs like in the similar areas of zeolites and MOFs.

Integrated innovation is simple in logical but is not easy in practise, which is same important to the innovation on new structure and/or new synthesis. Although the single step modification of reduction and ionization was developed 4-5 years ago (*Chem* 2018, 4, 1696; *J. Am. Chem. Soc.* 2019, 141, 14433), there is no integrated modification was reported because that not only a robust base COF is required but also the interference of multistep should be well avoided.

The synergistic effect of the ionic skeleton and C-N bonds for improving the electrocatalysis CO₂RR performance obviously possess the novelty. Thus, the CO₂RR electrocatalytic character of our COFs with C-N bonds and ionization skeleton is obviously the strong aspect of our work and therefore the novelty cannot be simply ignored.

On the other hand, our work is different to the reported single step post-modifications (*Chem* 2018, 4, 1696; *J. Am. Chem. Soc.* 2019, 141, 14433) in terms of materials, applications and / or synthesis. Deng *et al.* modified COF-300 by a direct reduction. However, the resultant COF-300 showed much lower FE_{CO} and current density than that of our CoTAPP-PATA-COFs (*Chem* 2018, 4, 1696-1709), which confirms the significance of multifunction of our COFs on the CO₂RR. In *J. Am. Chem. Soc.* 2019, 141, 14433, Guo *et al.* used Viologens (C₅H₄NR)₂ⁿ⁺ to construct a cationic radical framework from the 2,2'-bipyridine-based COF by a quaternization reaction. In comparison, we modified the CoTAPP-PATA-COF with CH₃I by a Menshutkin reaction. Therefore, the material, synthesis method and application of our work are different to the reference.

In addition, the CoTAPP-PATA-COF used in this work was firstly reported by our group in the

above-mentioned *Angew. Chem. Int. Ed.* 2022, 407, e202213522. As explained before, the CoTAPP-PATA-COF is previously reported as a bifunctional catalyst for ORR and OER. This work endows the existed COF with new applications, which is enlightening for the wide readership of *Nat Commun* to develop new functions on the existed numerous COFs.

It is true that the porphyrin based CO₂RR catalysts were widely studied because that the coordination of porphyrin and metal atom provide stable structure and high reactivity for the reaction. This is common for the catalysis research. For example, the catalysis of Fischer–Tropsch process is focusing on Fe, Co based catalysts in the past nearly 100 years, which is still attracting a lot of research with important progress. In this work, the reason of employing the porphyrin-based catalyst is controlling the variable for easily comparing our results with reference.

In addition, the manuscript has been revised thoughtfully, which is marked in the revision and summarized as followings.

✓ **Reorganization of Introduction section**

In the revision, the Introduction section was revised. All the references that the Reviewer mentioned were cited in the revision. In addition, the references were well reviewed and described in detail to explain the similarities and differences to our proposal.

“The bottom-up synthetic approach is the most common method to directly construct functional COFs.^[38-42] Challenged by the steric hindrance effect, the solubility difference and the microscopic reversibility, however, some functionalities cannot be introduced directly into COFs *via* the bottom-up syntheses.^[43-47] For the pre-designed COF that contains amine bonds and ionization skeletons, the existing covalent connection methods (*e.g.*, boroxine rings,^[48] imine bonds^[49] and β -ketoenamine linkage^[50]) do not support to obtain C-N linkage through the bottom-up synthesis directly, and the electrostatic repulsion effects of ionic building units impede the direct formation of ionization skeletons. Alternatively, the post-synthetic modification strategy provide a promising chance to construct new functional skeletons, pores, and linkages at the molecular level with controllable catalytic properties.^[51-56] Several single-step post-modifications were proposed to endow the base COFs with special properties.^[57-60] Deng *et al.* constructed imine-COFs by a post-reduction method to improve gas diffusion on electrode.^[61] Guo *et al.* used Viologens (C₅H₄NR)₂ⁿ⁺ to construct a cationic radical framework from the 2,2'-bipyridine-based COF, which showed high

photothermal conversion efficiencies.^[62] Thus, it is expectable to obtain the COF with reduced imine linkage and ionic skeleton if the single-step post-modification of reduction and ionization can be well integrated. However, the integrated multilevel post-modification is rarely reported, because that not only a robust base COF is required but also the interference of multistep should be well avoided. In the view of COF design, 4,4',4'',4'''-(1,4-phenylenebis(azanetriyl))tetrabenzaldehyde (PATA) unit contains ammonium groups which are potential to transfer to ionic skeleton, and 5,10,15,20-tetrakis(4-aminophenyl)porphinato]-cobalt (TAPP(Co)) unit can achieve efficient charge transfer due to their conjugated macrocyclic structures. In addition, we previously proved that CoTAPP-PATA-COF, composed by PATA and TAPP(Co), possesses good crystallinity, high surface areas, and excellent chemical stability.^[28] Thus, CoTAPP-PATA-COF is expected as a template for constructing multilevel post-synthetic modification COFs for CO₂RR.”

✓ **More structure and chemistry results**

In the revision, the thermal stability of these COFs was measured by the thermal gravimetric analysis (TGA) from 100 to 800 °C in N₂. As shown in Supplementary Fig. 22, the post-modified COFs kept the similar thermal stability to that of base COF, and no significant mass loss was observed at the temperature < 480 °C in N₂. This reveals that the thermal stability of COFs was well maintained after the post-modifications.

Supplementary Figure 22. TGA curves of CoTAPP-PATA-COF (black), N⁺-COF (green), NH-COF (blue) and N⁺-NH-COF (red) under N₂ atmosphere.

In the revision, the hydrophobic behavior of N⁺-NH-COF was a further verified by water uptake

test. As shown in Supplementary Fig. 24, the N⁺-NH-COF exhibited a low water vapour uptake capacity of 188 cm³ g⁻¹ at 298 K and an inflection point at high relative pressures (P/P₀ = 0.5), which are in good agreement with the hydrophobic behaviors (*Adv. Mater.* **2018**, *30*, 1704304). For the electrocatalysis of CO₂RR, the hydrophobic nature of catalyst is crucial to protect the active sites through suppressing the competitive adsorption of water, leading to the improvements of selectivity and energy efficiency (*Adv. Mater.* **2022**, *34*, 2205186).

Supplementary Figure 24. The water uptakes of N⁺-NH-COF at 298 K.

In the revision, the N⁺-NH-COF samples before and after the CO₂RR test were measured by XPS. As shown in Supplementary Fig. 40, the XPS I 3d spectrum of the fresh N⁺-NH-COF indicates I 3d_{5/2} peak at 630.5 eV and I 3d_{3/2} at 619.0 eV, assigned to iodide ions state (Ref.). For the used N⁺-NH-COF sample, no visible changes are observed, suggesting that I⁻ state is maintained after CO₂RR.

Supplementary Figure 40. The XPS I 3d spectra of N⁺-NH-COF before and after the CO₂RR.

In the revision, the FTIR spectra of CoTAPP-PATA-COF, N⁺-COF, NH-COF and N⁺-NH-COF

were remeasured. In addition, the solid-state ^{13}C NMR was employed to measure the carbon states in the COFs. Compared with the CoTAPP-PATA-COF, as shown in Supplementary Fig. 1, the C=N linkages at 1622 cm^{-1} were retained, while a new peak raised at 1470 cm^{-1} , being ascribed to $-\text{N}^+(\text{CH}_3)_2^-$ in the FTIR spectrum of ionized N^+ -COF (*Adv. Mater.* **34**, 2110496 (2022); *Small* **17**, 2004933 (2021)). This reveals that the skeleton ionization was solely realized by the Menshutkin reaction. After the reduction modification, in contrast, the FTIR spectra of NH-COF and the N^+ -NH-COF revealed that the C=N vibrations at 1622 cm^{-1} were totally replaced by the new signals of C-N at 1157 cm^{-1} (*Chem 2018*, *4*, 1696-1709). This indicates that the C=N linkages were fully converted into C-N bonds. Apart from C-N vibrations, furthermore, the ionic $-\text{N}^+(\text{CH}_3)_2^-$ was also detected for the FTIR spectrum of N^+ -NH-COF. Moreover, the Co-N bonds at 660 cm^{-1} were observed in all four COFs, indicating that the post-modifications were harmless to the Co-N coordination.

Supplementary Figure 1. The FTIR spectra of CoTAPP-PATA-COF (black), N^+ -COF (green), NH-COF (blue), N^+ -NH-COF (red), PATA monomer (yellow) and CoTAPP monomer (purple).

The functionalization of COFs was further confirmed by using the solid ^{13}C cross-polarization/magic-angle spinning solid-state nuclear magnetic resonance (CP/MAS ssNMR, Supplementary Fig. 2). The C=N signals (148 ppm) in CoTAPP-PATA-COF and N^+ -COF were completely replaced by the new C-N signals at 51 ppm for the NH-COF and N^+ -NH-COF. In addition, N^+ -COF and N^+ -NH-COF exhibited a new peak belonging to methyl groups at 60 ppm. These results indicate that the successful reduction of $-\text{C}=\text{N}$ and the methylation with charge state changes.

Supplementary Figure 2. The solid ^{13}C NMR spectra of CoTAPP-PATA-COF (black), N^+ -COF (green), NH-COF (blue) and N^+ -NH-COF (red).

In the revision, the XPS results of the COFs with higher quality were obtained, as shown in the new Figure 3A. As shown in Figure 3A, the high-resolution Co 2p spectra of the four COFs exhibited Co-N coordination, which confirmed that the Co-N sites were well retained after the multilevel post-modifications. In detail, compared with CoTAPP-PATA-COF (781.38 eV) and NH-COF (781.36 eV), the Co $2p_{3/2}$ spectra of N^+ -COF (780.92 eV) and N^+ -NH-COF (780.90 eV) showed a negative shift of ~ 0.45 eV, which is ascribed to the electron-withdrawing effect of methyl groups. This proves that the skeletons in N^+ -COF and N^+ -NH-COF have been ionized by the CH_3I modification.

Figure 3A. The high-resolution Co 2p spectra of the four COFs.

✓ More results on reactivity of modified COF

In the revision, the electronic conductivity of the CoTAPP-PATA-COF, N⁺-COF, NH-COF and N⁺-NH-COF were measured by the four-probe method at 298 K. As shown in Supplementary Fig. 26, N⁺-NH-COF and N⁺-COF have the similar electronic conductivity of $6.7 \times 10^{-9} \text{ S m}^{-1}$ and $8.1 \times 10^{-9} \text{ S m}^{-1}$, respectively, which are one order of magnitude larger than those of CoTAPP-PATA-COF ($8.5 \times 10^{-10} \text{ S m}^{-1}$) and NH-COF ($3.0 \times 10^{-10} \text{ S m}^{-1}$). It suggests that the ionized skeletons can promote the electron transfer along the frameworks, and thus improving the activity.

Supplementary Figure 26. The I-V curves of (A) CoTAPP-PATA-COF, (B) NH-COF, (C) N⁺-COF and (D) N⁺-NH-COF by the four-probe measurement at 298 K.

In the revision, the Nyquist plots of these COFs were measured. As shown in Supplementary Fig. 34, the charge transfer resistances (R_{ct}) over N⁺-COF and N⁺-NH-COF were 36 and 24 Ω , respectively, being smaller than that of CoTAPP-PATA-COF (52 Ω) and NH-COF (42 Ω). This suggested that the ionization of skeletons enhanced the charge transfer capacity of COFs.

Supplementary Figure 34. EIS spectra of CoTAPP-PATA-COF (black), N⁺-COF (green), NH-COF (blue) and N⁺-NH-COF (red) (insert: typical equivalent circuit).

In the revision, the electrochemically active surface areas of these COFs were evaluated for the comparison of the exposed active sites. As shown in Figure 4F, the electrochemical double layer capacitances (C_{dl}) for the CoTAPP-PATA-COF, N⁺-COF, NH-COF, and N⁺-NH-COF were 16.92, 26.34, 24.37, and 28.76 mF cm⁻², respectively (Figure 4F). To investigate the exposed active sites, the ECSA of the COFs were then calculated by $ECSA = C_{dl}/C_s$, where the C_s is 0.04 mF cm⁻² (*CCS Chem.*, **2022**, 1-14). N⁺-NH-COF offered the highest ECSA (719) relative to CoTAPP-PATA-COF (423), N⁺-COF (659), and NH-COF (609), which is in line with its superior CO₂RR performance. Moreover, the ECSA of N⁺-COF was higher than the base COF and NH-COF, suggesting that the ionic frameworks provided more active sites.

Figure 4F. The C_{dl} values of CoTAPP-PATA-COF (black), N⁺-COF (green), NH-COF (blue) and N⁺-NH-COF (red).

In the revision, the normalized current densities on the COFs, *i.e.*, j_{CO} per ECSA, were further

evaluated by using j_{CO} and ECSA. As shown in Figure 4G, within the potential range of -0.5 to -1.0 V, the ionized COFs of N^+ -COF and N^+ -NH-COF always exhibited the higher normalized current density than that of un-ionized COFs of CoTAPP-PATA-COF and NH-COF, confirming that the ionization helps to improve the COF activity towards CO_2RR .

Figure 4G. The current density of ECSA for CoTAPP-PATA-COF (black), N^+ -COF (green), NH-COF (blue) and N^+ -NH-COF (red).

In the revision, the role of C-N bonds on CO_2 adsorption was further checked by the CO_2 -temperature programmed desorption (CO_2 -TPD). Specifically, NH-COF exhibited a higher CO_2 desorption signal than that of base CoTAPP-PATA-COF, indicating the enhanced CO_2 adsorption capacity on C-N bonds (Supplementary Fig. 8).

Supplementary Figure 8. The CO_2 -TPD curves of CoTAPP-PATA-COF (black) and NH-COF (blue).

✓ Improved theoretical calculations

In the revision, we reinvestigated the catalytic process by considering the suggestion of reviewer on the effects of charge states. When we take the counterions (Γ) into consideration, the energy differences between the intermediates become much larger as shown in the new Figure 5B. More interestingly, the introduced counterions makes lower potential energy surface for M-I-N⁺-COF and M-I-N⁺-NH-COF, which may be a suggestion that the counterions is favorable for the CO₂RR.

Figure 5. (A) The optimized geometrical structures and (B) Calculated free energy diagram of M-CoTAPP-PATA-COF, M-I-N⁺-COF, M-NH-COF and M-I-N⁺-NH-COF catalyzing CO₂RR (M-Model).

As shown in Figure 5B, CO₂RR includes three steps: the formation of *COOH and *CO where a proton coupled a single electron transfer take place for each step, and CO desorption from the active metal center. From CO₂ to the product CO, the CO₂RR catalyzed by M-I-N⁺-COF and M-I-N⁺-NH-COF are slightly endergonic, while those catalyzed by M-CoTAPP-PATA-COF and M-NH-COF are exergonic. This is in accordance with the excellent catalytic ability of the four COFs. The relative free energies of *COOH formation are higher than the initial state of CO₂ (*i.e.*, the relative zero point of free energies), while the process from *COOH to the final product CO are largely exergonic.

This indicates that the formation of *COOH is the rate control step. The lowest free energy of *COOH state for M-I-N⁺-NH-COF is in line with its best performance from a thermodynamic viewpoint. In addition, both M-I-N⁺-COF and M-I-N⁺-NH-COF have the lower free energies of *COOH than those of M-CoTAPP-PATA-COF and M-NH-COF, which indicates a stronger promotion effect of ionization than the neutral ones.

Therefore, i) the relative free energies of *COOH formation are higher than the initial state of CO₂, and thus ii) the formation of *COOH is the rate-determining step, iii) the energy change from CO₂ to *CO in the revised calculation is ~1.0 eV, in line with the reported values (e.g., *J. Am. Chem. Soc.* **2021**, *143*, 18052–18060).

In summary, the new Scheme 1, Figure 3A, Figure 4G and Figure 5A&B, and additional 10 Supplementary Figures were provided in the revision, which improve the quality of this manuscript.

Reviewer #3:

In this manuscript, the authors demonstrate the utilization of the double modifications of the CoTAPP-PATA-COF for the regulation of selectivity and activity of the electrocatalytic CO₂ reduction reactions. The linkages and linker have been respectively modified by reduction and ionization reactions. The authors have used FTIR and XRD to characterize the synthesis of the pristine COF and the COFs after post-modifications. The results showed that the post-synthetic modification strategy can adjust the performance of CO₂RR. Notably, the dual-functional COF achieved high selectivity with a maximum CO Faradaic efficiency of 97.32% and high turnover frequencies of 9922.68 h⁻¹. Despite this, the manuscript is still short of a few fundamental aspects regarding the structural characterizations and interpretation of the mechanism for the observation of modified performance. The reviewer suggests that this manuscript is too premature to be published before the following issues get properly addressed.

=====

Response:

We're very thankful for the highly valuable and insightful questions, as well as the constructive suggestions from Reviewer #3, which have certainly helped us to improve the quality of this manuscript. The manuscript was carefully revised based on these suggestions. Additional experiments and reasonable analysis were performed to provide solid supports in the revision.

For clarity and quick understanding, the corresponding results for answering the questions raised in above general comment were summarized and guided, firstly.

To address the suggestion of "*the structural characterizations*", additional characterizations (*e.g.*, FTIR, ¹³C NMR) were conducted to support the COF structure before and after post-modifications (please see details in **Response to Comment #2**).

To address the suggestion of "*interpretation of the mechanism*", the electronic conductivity, EIS, ECSA and the normalized current density were measured, which support that the ionic skeletons can improve the activity and the reduced C-N bonds can enhance the CO₂ adsorption (please see details in **Response to Comment #1**). In addition, the theoretical calculations of the reaction mechanism on the modified COFs were improved in the revision (please see details in **Response to Comment #7**).

=====

1. The authors have given a few assumptions in the whole manuscript. These assumptions are not exactly consistent which makes it hard for the reader to understand which is essentially the

underlying mechanism for the improved performance. For example, in the abstract, the authors demonstrate “Amine-linkages favour molecular diffusion, and the ionization skeletons enable to delocalize charges to construct a protected ion transport pathway”. In lines 130-132, the authors indicate that the ionization for the skeleton can disturb the interlayer stacking because of the electrostatic repulsion and steric hindrance, and promote the exposure of active sites and provide more space for the electrochemical reaction. In line 197, the authors suggested ionization units could improve the activity and the C-N bonds could promote CO₂ adsorption. The authors need to carefully articulate their key hypothesis.

Response:

We appreciate this comment.

In the revision, the key hypothesis was carefully articulated as that the ionized skeletons improve the electronic conductivity along the frameworks, and the reduced amine-linkages enhanced the adsorption capacity of CO₂. To support the hypothesis, additional conclusive experiments (e.g., electronic conductivity, EIS, ECSA, normalized current density, and CO₂-TPD) have been carried out yielding supportive results, which were discussed in detail as the followings.

Supplementary Figure 26. The I-V curves of (A) CoTAPP-PATA-COF, (B) NH-COF, (C) N⁺-COF and (D) N⁺-NH-COF by the four-probe measurement at 298 K.

Electronic conductivity

In the revision, the electronic conductivity of the CoTAPP-PATA-COF, N⁺-COF, NH-COF and N⁺-NH-COF were measured by the four-probe method at 298 K. As shown in Supplementary Fig. 26, N⁺-NH-COF and N⁺-COF have the similar electronic conductivity of $6.7 \times 10^{-9} \text{ S m}^{-1}$ and $8.1 \times 10^{-9} \text{ S m}^{-1}$, respectively, which are one order of magnitude larger than those of CoTAPP-PATA-COF ($8.5 \times 10^{-10} \text{ S m}^{-1}$) and NH-COF ($3.0 \times 10^{-10} \text{ S m}^{-1}$). It suggests that the ionized skeletons can promote the electron transfer along the frameworks, and thus improving the activity.

Charge transfer resistances

In the revision, the Nyquist plots of these COFs were measured. As shown in Supplementary Fig. 34, the charge transfer resistances (R_{ct}) over N⁺-COF and N⁺-NH-COF were 36 and 24 Ω , respectively, being smaller than that of CoTAPP-PATA-COF (52 Ω) and NH-COF (42 Ω). This suggested that the ionization of skeletons enhanced the charge transfer capacity of COFs.

Supplementary Figure 34. EIS spectra of CoTAPP-PATA-COF (black), N⁺-COF (green), NH-COF (blue) and N⁺-NH-COF (red) (insert: typical equivalent circuit).

Electrochemically active surface areas

In the revision, the electrochemically active surface areas of these COFs were evaluated for the comparison of the exposed active sites.

As shown in Figure 4F, the electrochemical double layer capacitances (C_{dl}) for the CoTAPP-PATA-COF, N⁺-COF, NH-COF, and N⁺-NH-COF were 16.92, 26.34, 24.37, and 28.76 mF cm^{-2} , respectively (Figure 4F). To investigate the exposed active sites, the ECSA of the COFs were then calculated by $\text{ECSA} = C_{dl}/C_s$, where the C_s is 0.04 mF cm^{-2} (*CCS Chem.*, **2022**, 1-14). N⁺-NH-COF offered the highest ECSA (719) relative to CoTAPP-PATA-COF (423), N⁺-COF (659), and NH-

COF (609), which is in line with its superior CO₂RR performance. Moreover, the ECSA of N⁺-COF was higher than the base COF and NH-COF, suggesting that the ionic frameworks provided more active sites.

Figure 4F. The C_{dl} values of CoTAPP-PATA-COF (black), N⁺-COF (green), NH-COF (blue) and N⁺-NH-COF (red).

Normalized current density

In the revision, the normalized current densities on the COFs, *i.e.*, j_{CO} per ECSA, were further evaluated by using j_{CO} and ECSA. As shown in Figure 4G, within the potential range of -0.5 to -1.0 V, the ionized COFs of N⁺-COF and N⁺-NH-COF always exhibited the higher normalized current density than that of un-ionized COFs of CoTAPP-PATA-COF and NH-COF, confirming that the ionization helps to improve the COF activity towards CO₂RR.

Figure 4G. The j_{CO} normalized by ECSA for CoTAPP-PATA-COF (black), N⁺-COF (green), NH-COF (blue) and N⁺-NH-COF (red).

Interactions between CO₂ and COFs

In the revision, the role of C-N bonds on CO₂ adsorption was further checked by the CO₂-temperature programmed desorption (CO₂-TPD). Specifically, NH-COF exhibited a higher CO₂ desorption signal than that of base CoTAPP-PATA-COF, indicating the enhanced CO₂ adsorption capacity on C-N bonds (Supplementary Fig. 8).

Supplementary Figure 8. The CO₂-TPD curves of CoTAPP-PATA-COF (black) and NH-COF (blue).

All above results were provided in the Supplementary Materials and / or the Main Text. These results were discussed in the manuscript to support the conclusion that the ionized skeletons improve the electronic conductivity along the frameworks, and the reduced amine-linkages enhanced the adsorption capacity of CO₂. Correspondingly, the mistakes that the reviewer mentioned in Abstract, Line 130-132, and Line 197 have been corrected in the revision.

=====

2. Further evidence is needed for the implementation of the dual-level modification.

a) The authors indicated in Figure S1 that -C=N was weakened by reduction, and it was unreasonable to attribute the remaining peak to -C=N in CoTAPP. The peak of pyrrole -C=N FTIR in porphyrins was usually 1340cm⁻¹, and the author needed to indicate the position of -C=N in the IR spectrum. Solid-state ¹³C NMR could be a good technique to further support whether -C=N is successfully reduced to -C-N.

=====

Response:

We appreciate this comment.

In the revision, the FTIR spectra of CoTAPP-PATA-COF, N⁺-COF, NH-COF and N⁺-NH-COF were remeasured. In addition, the solid-state ¹³C NMR was employed to measure the carbon states in the COFs.

Compared with the CoTAPP-PATA-COF, as shown in Supplementary Fig. 1, the C=N linkages at 1622 cm⁻¹ were retained, while a new peak raised at 1470 cm⁻¹, being ascribed to -N⁺-(CH₃)₂⁻ in the FTIR spectrum of ionized N⁺-COF (*Adv. Mater.* **34**, 2110496 (2022); *Small* **17**, 2004933 (2021)). This reveals that the skeleton ionization was solely realized by the Menshutkin reaction. After the reduction modification, in contrast, the FTIR spectra of NH-COF and the N⁺-NH-COF revealed that the C=N vibrations at 1622 cm⁻¹ were totally replaced by the new signals of C-N at 1157 cm⁻¹ (*Chem* *2018*, *4*, 1696-1709). This indicates that the C=N linkages were fully converted into C-N bonds. Apart from C-N vibrations, furthermore, the ionic -N⁺-(CH₃)₂⁻ was also detected for the FTIR spectrum of N⁺-NH-COF. Moreover, the Co-N bonds at 660 cm⁻¹ were observed in all four COFs, indicating that the post-modifications were harmless to the Co-N coordination.

Supplementary Figure 1. The FTIR spectra of CoTAPP-PATA-COF (black), N⁺-COF (green), NH-COF (blue), N⁺-NH-COF (red), PATA monomer (yellow) and CoTAPP monomer (purple).

The functionalization of COFs was further confirmed by using the solid ¹³C cross-polarization/magic-angle spinning solid-state nuclear magnetic resonance (CP/MAS ssNMR, Supplementary Fig. 2). The C=N signals (148 ppm) in CoTAPP-PATA-COF and N⁺-COF were completely replaced by the new C-N signals at 51 ppm for the NH-COF and N⁺-NH-COF. In addition, N⁺-COF and N⁺-NH-COF exhibited a new peak belonging to methyl groups at 60 ppm. These results indicate that the successful reduction of -C=N and the methylation with charge state

changes.

The above results were provided in the Supplementary Materials and discussed in the Main Text.

Supplementary Figure 2. The solid ^{13}C NMR spectra of CoTAPP-PATA-COF (black), N^+ -COF (green), NH-COF (blue) and N^+ -NH-COF (red).

2 b) Given that the methylation was done with a large excess of CH_3I , the readers would wonder why only the N atoms in PATAN were methylated instead of N atoms in imine bonds.

Response:

With the Menshutkin reaction, the secondary amine groups are more easily methylated compared with the C=N bonds, which is further supported by the previous work that only C-N bonds rather than the co-existed C=N bonds in COF were methylated through the Menshutkin reaction (*ACS Sustainable Chem. Eng.* **2022**, *10*, 9749-9759).

In the revision, the selective methylation of C-N was further confirmed by the solid-state ^{13}C NMR and FTIR results of N^+ -COF, which show the unchanged C=N signals and the new generated methylation signals (please see details from **Response to Comment #3a**).

The key point of above explanation was provided in the Main Text.

3. The authors proved the adsorption and binding of CO_2 by post-modification, but the actual CO_2 adsorption value measured is contrary to the author's statement. The author defined that the adsorption of CO_2 by different COF is unreasonable by $n = \text{SCO}_2/\text{SBET}$. This is because the performance of the electrocatalyst CO_2RR is tested using catalysts with the same mass rather than the same specific surface areas.

Response:

We appreciate this comment.

The reviewer is right that the catalysts used in CO₂RR were accounted by the same mass rather than the same specific surface areas. Therefore, we removed the unreasonable explanation of $n=S_{CO_2}/S_{BET}$ in the revision.

On the other hand, the CO₂-TPD experiments were carried out to prove the adsorption and binding capacity of CO₂ on the reduced COF. Please see details from the **Response to Comment #1**.

The corresponding results were provided in the Supplementary Materials and discussed in the Main Text.

4. XPS analysis:

a) The signal-to-noise ratio of XPS data (Figure 3a) for Co 2p is too low to do any convincing interpretation. The fitting of Co 2p is very poor as well.

b) The author stated that Co 2p showed about 0.13 eV blue shift due to the electron-withdrawing effect, however, it was exactly the opposite of the data as shown in Figure 3a. It seems that there is a redshift after N is methylated.

Response:

We appreciate this comment.

In the revision, the XPS results of the COFs with higher quality were obtained, as shown in the new Figure 3A.

As shown in Figure 3A, the high-resolution Co 2p spectra of the four COFs exhibited Co-N coordination, which confirmed that the Co-N sites were well retained after the multilevel post-modifications. In detail, compared with CoTAPP-PATA-COF (781.38 eV) and NH-COF (781.36 eV), the Co 2p_{3/2} spectra of N⁺-COF (780.92 eV) and N⁺-NH-COF (780.90 eV) showed a negative shift of ~0.45 eV, which is ascribed to the electron-withdrawing effect of methyl groups. This proves that the skeletons in N⁺-COF and N⁺-NH-COF have been ionized by the CH₃I modification.

Figure 3A. The high-resolution Co 2p spectra of the four COFs.

The above results were provided and discussed in the Main Text.

5. The conductivity of the material can not solely determined by the band gap, and it needs to be independently determined by conductivity measurement.

Response:

We appreciate this comment.

In the revision, the conductivity of the COFs was further measured by the four-probe method at 298 K. As shown in Supplementary Fig. 27, N⁺-NH-COF and N⁺-COF have the similar electronic conductivity of $6.7 \times 10^{-9} \text{ S m}^{-1}$ and $8.1 \times 10^{-9} \text{ S m}^{-1}$, respectively, which are one order of magnitude larger than those of CoTAPP-PATA-COF ($8.5 \times 10^{-10} \text{ S m}^{-1}$) and NH-COF ($3.0 \times 10^{-10} \text{ S m}^{-1}$). It suggests that the ionized skeletons can promote the electron transfer along the frameworks, and thus improving the activity. Please see details in **Response to Comment #1**.

The related results were provided in the Supplementary Materials and discussed in Main Text.

6. The electrocatalytic performance presented by the author in the manuscript diagram is inconsistent with the performance described in the manuscript, which needs to be further confirmed.
 a) The authors suggest that -C-N linkages improved the catalytic activity and the ionic linker contributed to higher selectivity, contrary to the results in Figure 4c, where -C-N linkages appear to

increase selectivity and ionic linkers appear to increase activity.

Response:

We appreciate this comment.

We have corrected the explanation of the effects of C-N linkages and the ionic linkers on the CO₂RR catalytic performance, respectively. To support the conclusions, additional conclusive experiments (e.g., electronic conductivity, EIS, ECSA, normalized current density, and CO₂-TPD) have been carried out yielding supportive results, which were discussed in detail in **Response to Comment #1**.

6 b). ECSA data show that NH-COF > N⁺-NH-COF, which is opposite to the reactivity shown in Figure 4d, please confirm.

Response:

We appreciate this comment.

Figure 4F. The C_{dl} values of CoTAPP-PATA-COF, N⁺-COF, NH-COF and N⁺-NH-COF.

In the revision, the C_{dl} values for the CoTAPP-PATA-COF, N⁺-COF, NH-COF, and N⁺-NH-COF were confirmed as 16.92, 26.34, 24.37, and 28.76 mF cm⁻², respectively (Figure 4F). To investigate the exposed active sites, the electrochemically active surface areas (ECSAs) of the COFs were then calculated by $ECSA = C_{dl}/C_s$, where the C_s is 0.04 mF cm⁻². N⁺-NH-COF offered the highest ECSA (719) relative to CoTAPP-PATA-COF (423), N⁺-COF (659), and NH-COF (609), which is in line with its CO₂RR performance. Moreover, the ECSA of N⁺-COF was higher than the base COF and

NH-COF, suggesting that the ionic frameworks provided more active sites.

These results were provided and discussed in the Main Text.

7. Theoretical calculations need to be further confirmed.

a) The authors omitted many important details of the DFT calculation part, which can affect its reliability. The authors should clearly mention that the calculations were done based on the molecular models, which are hydrogen-terminated fragments of the COFs, instead of using the COFs themselves. Correspondingly, the naming of the materials in Figure 5a should be revised. The authors should explain in the manuscript why the use of such molecular models is reasonable and point out how this simplification may affect the results. Since N⁺-COF and N⁺-NH-COF are positively charged, in the calculations, how are the corresponding counterions of the positively charged species treated? It is also not known if the authors have incorporated the thermodynamic corrections in their calculation.

Response:

We appreciate this comment.

The DFT calculations were carefully revised.

In the revision, the DFT calculations were carried out at the theoretical level of CAM-B3LYP/6-311G(d) (SDD for Co) for the cluster models of these four COFs. Reasonable geometrical, electronic and thermodynamic information could be obtained with cluster models for COFs and similar systems. All structures along the potential energy surfaces were optimized without any restrictions. Frequencies were further performed to confirm that all optimized geometries are local minima and to obtain the Gibbs free energies. In the calculation, the influence of corresponding counterions (I⁻) was considered and the four model molecules were labelled as M-CoTAPP-PATA-COF, M-I-N⁺-COF, M-NH-COF and M-I-N⁺-NH-COF, respectively, where the counter ion I⁻ was considered for the N⁺-COF and N⁺-NH-COF (Figure 5A).

Above information was described in the Main Text.

7 b) The reviewer noticed that in the energy profile (Figure 5a), the free energy differences for the same kind of intermediate from different catalysts are very small. For example, the energy difference of COOH* mediated by the types of catalysts is smaller than 0.04 eV. Such small energy differences, which may be within the error of the theoretical calculations, cannot be used as support to correlate with the experimental results of the catalysts.

c) Is the calculation of free energy in Figure 5A in the manuscript reasonable? i) The whole energy

profile is going down which makes the reader wonder that CO₂RR is a spontaneous reaction; ii) For Figure 5A, it seems the formation of the CO* is the rate-determining step? iii) The free energy change from CO₂ to CO* is too large than normal. The reviewer suggests that the authors carefully check their calculation method.

Response:

We're very thankful for this comment.

In the revision, we reinvestigated the catalytic process by considering the suggestion of reviewer on the effects of charge states. When we take the counterions (Γ) into consideration, the energy differences between the intermediates become much larger as shown in the new Figure 5B. More interestingly, the introduced counterions makes lower potential energy surface for M-I-N⁺-COF and M-I-N⁺-NH-COF, which may be a suggestion that the counterions is favorable for the CO₂RR.

Figure 5. (A) The optimized geometrical structures and (B) Calculated free energy diagram of M-CoTAPP-PATA-COF, M-I-N⁺-COF, M-NH-COF and M-I-N⁺-NH-COF catalyzing CO₂RR (M-Model).

As shown in Figure 5B, CO₂RR includes three steps: the formation of *COOH and *CO where a proton coupled a single electron transfer take place for each step, and CO desorption from the active

metal center. From CO₂ to the product CO, the CO₂RR catalyzed by M-I-N⁺-COF and M-I-N⁺-NH-COF are slightly endergonic, while those catalyzed by M-CoTAPP-PATA-COF and M-NH-COF are exergonic. This is in accordance with the excellent catalytic ability of the four COFs. The relative free energies of *COOH formation are higher than the initial state of CO₂ (*i.e.*, the relative zero point of free energies), while the process from *COOH to the final product CO are largely exergonic. This indicates that the formation of *COOH is the rate control step. The lowest free energy of *COOH state for M-I-N⁺-NH-COF is in line with its best performance from a thermodynamic viewpoint. In addition, both M-I-N⁺-COF and M-I-N⁺-NH-COF have the lower free energies of *COOH than those of M-CoTAPP-PATA-COF and M-NH-COF, which indicates a stronger promotion effect of ionization than the neutral ones.

Therefore, i) the relative free energies of *COOH formation are higher than the initial state of CO₂, and thus ii) the formation of *COOH is the rate-determining step, iii) the energy change from CO₂ to *CO in the revised calculation is ~1.0 eV, in line with the reported values (e.g., *J. Am. Chem. Soc.* **2021**, *143*, 18052–18060).

The above results were discussed in detail in the Main Text.

=====
8. The whole manuscript needs to be checked carefully. There are many mismatches between the material name and the corresponding data in the article, which will cause misunderstandings among the readers. For example, supplementary Figure S25 and Supplementary Figure S27 are both for N⁺-COF. One of them must be incorrectly labeled. The red line in the Figure S30, which is for FE_{co} is in the exact same shape as the black line. The FE data is wrongly plotted.
=====

Response:

We appreciate this comment.

In the revision, the caption of Figure S27 (*i.e.*, Supplementary Figure 32 in the revision) has been corrected. At the same time, Figure S30 (*i.e.*, Supplementary Figure 35 in the revision) was also revised.

In addition, the manuscript was checked and revised carefully, which was mark in blue in the revision.

Supplementary Figure 32. Chronoamperometric responses of N^+ -NH-COF at -0.5 (red), -0.6 (blue), -0.7 (green), -0.8 (pink), -0.9 (cyan), and -1.0 V (brown) (vs. RHE).

Supplementary Figure 35. Chronoamperometry test for CO_2RR of N^+ -NH-COF at a potential of -0.8 V in 0.5 M $KHCO_3$ under CO_2 atmosphere.

9. Other issues:

- a) TAPP and PATA, as simplified names, need to be defined the first time they appear in the manuscript.
- b) Line 127-128: “Their highly crystal structures...” should be “Their highly crystalline structures...”.
- c) In lines 228-229, “the NH-COF showed the highest Tafel slopes than that of other COFs, indicating the superior combination with CO_2 in the process of CO_2RR ” reads confusing. The highest Tafel slopes indicate that the kinetics are most unfavorable.
- d) In Supplementary Table S5, the standard for the comparison of performance parameters is not uniform. For example, the author gives FE (97.37%) when the potential is -0.8 V, but J_{CO} (21.37) is given when the potential -0.9 V for the same material (N^+ -NH-COF).
- e) In the abstract “OOH*” intermediates should be “COOH*”.

Response:

We appreciate this comment.

(a) The full names of PATA and TAPP were provided in the first time they appeared in the manuscript.

(b) It has been corrected in the revision.

(c) For clarity, the related content has been revised as “The corresponding Tafel slope of the CoTAPP-PATA-COF was 236 mV dec⁻¹, which declined to 203, 197, and 184 mV dec⁻¹ for N⁺-COF, NH-COF, and N⁺-NH-COF, respectively (Figure 4B). It suggests that the post modifications significantly improve the electrocatalytic CO₂RR kinetics.” in the revision.

(d) For clarity, the FE_{CO} at -0.8 V and -1.0 V were listed and compared. In addition, the TOF and j_{CO} values at -1.0 V were compared in the revision.

Supplementary Table 5. Summary of recently reported CO₂RR performances of other reported COF derived electrocatalysts under alkaline conditions (Electrolyte 0.5 M KHCO₃).

Electrocatalyst	FE _{CO} (%)	FE _{CO} (%)	TOF (s ⁻¹)	j _{CO} (mA cm ⁻²)	Reference
	at -0.8 V	at -1.0 V	at -1.0 V	at -1.0 V	
CoTAPP-PATA-COF	81	67	1.08	15.3	This work
N ⁺ -COF	95	78	1.86	18.1	
NH-COF	94	83	1.74	16.5	
N ⁺ -NH-COF	97	83	2.48	21.4	
CoPc-PI-COF-1	95	~82	4.90	21.2	[S2]
COF-300-AR	80	-	-	-	[S3]
NiPc-COF	93	~94	1.05	35.0	[S4]
COF-367-Co	91	~85	0.50	33.0	[S5]
CoP-BDT _{HexO} -COF	98	~90	2.40	10.8	[S6]
TAPP(Co)-B18C6-COF	93	71	0.35	9.5	[S7]
Co-TTCOF	~88	-	~1.10	~2.5	[S8]
TT-Por(Co)-COF	~87	-	~0.10	~5.6	[S9]

(e) It has been corrected in the revision.

Moreover, we checked and revised the manuscript and the Supplementary Materials carefully and thoroughly in the revision, which were marked in the revised files.

REVIEWER COMMENTS

Reviewer #1 (Remarks to the Author):

The authors have been addressed the issues, now it can be published in NC.

Reviewer #3 (Remarks to the Author):

The reviewers addressed most of the reviewer's concern.

However, there is still issue with their DFT calculations. The authors did not calculate the free energy correctly. If done properly, all the $*+CO+H_2O$ species should have the exact same values because the free energy here is a relative value. The reviewer suggests the authors check their methods carefully. The reviewer guess that this is probably a reason why differences for the free energies of $*COOH$ are so small.

In addition, Figure 3 is titled "Chemical structure of four COFs." which is not correct. Figure 3 doesn't give any information for chemical structure, and instead it is about electronic characterizations.

Reply to Referees' comments

Reply to Referee #3

Reviewer #3: The reviewers addressed most of the reviewer's concern.

=====

1. However, there is still issue with their DFT calculations. The authors did not calculate the free energy correctly. If done properly, all the $*+CO+H_2O$ species should have the exact same values because the free energy here is a relative value. The reviewer suggests the authors check their methods carefully. The reviewer guess that this is probably a reason why differences for the free energies of $*COOH$ are so small.

=====

Thank you for your comments.

We are so sorry to make a mistake to describe the free energy changes in the catalytic process. We have revised the name of the ordinate to " ΔG " from the "free energy" in Figure 5B.

Figure 5B. The calculated free energy change (ΔG) diagram of M-CoTAPP-PATA-COF, M-I-N⁺-COF, M-NH-COF and M-I-N⁺-NH-COF catalyzing CO₂RR (M-Model).

These are the equations of the free energy (G) for different steps in the calculation method:

- (1) $*+CO_2 \rightarrow *CO_2$ $G_1 = G_{*CO_2} - G^* - G_{CO_2}$
- (2) $*+CO_2 + H^+ + e^- \rightarrow *COOH$ $G_2 = G_{*COOH} - G^* - G_{CO_2} - G_{H^+} - eU$
- (3) $*+CO_2 + 2H^+ + 2e^- \rightarrow *CO + H_2O$ $G_3 = G_{*CO} + G_{H_2O} - G^* - G_{CO_2} - 2G_{H^+} - 2eU$
- (4) $*+CO_2 + 2H^+ + 2e^- \rightarrow *+CO + H_2O$ $G_4 = G^* + G_{CO} + G_{H_2O} - G^* - G_{CO_2} - 2G_{H^+} - 2eU$

And the free energy changes (ΔG) values of every step (ΔG_2 , ΔG_3 and ΔG_4) were calculated from the value of G_2-G_1 , G_3-G_2 and G_4-G_3 (*J. Am. Chem. Soc.* **2021**, *143*, 18052–18060). Thus, the free energy changes value (ΔG_4) of the $^*+CO+H_2O$ species have the different values.

To better describe the relative free energies in catalytic process, we have added the diagram of the calculated free energy value (G) as shown in Supplementary Fig. 41. Accordingly, the $^*+CO_2$ (CO_2 adsorption) was set as the background value (the relative zero point of free energies) (*Angew. Chem. Int. Ed.* **2021**, *60*, 4864 and *Chem. Eng. J.* **2022**, *450*, 138427). Thus, the free energy value (G_4) of the $^*+CO+H_2O$ species have the exact same values.

Supplementary Fig. 41. Calculated free energy diagram of M-CoTAPP-PATA-COF, M-I-N+-COF, M-NH-COF and M-I-N+-NH-COF catalyzing CO_2RR (M-Model).

Table R1. The values of G values (eV) in the catalytic process.

	M-I-N ⁺ -NH-COF	M-I-N ⁺ -COF	M-NH-COF	M-CoTAPP-PATA-COF
	0.000	0.000	0.000	0.000
G_1	-0.995	-1.863	-0.709	-0.715
G_2	2.044	1.285	2.482	2.489
G_3	3.057	2.143	3.288	3.288
G_4	3.083	3.083	3.083	3.083

Table R2. The corresponding ΔG values (eV) in the catalytic process.

	M-I-N ⁺ -NH-COF	M-I-N ⁺ -COF	M-NH-COF	M-CoTAPP-PATA-COF
ΔG_2	3.039	3.147	3.191	3.204
ΔG_3	1.013	0.858	0.807	0.800
ΔG_4	0.025	0.940	-0.206	-0.206

We have revised the figure 5B, updated the supporting information of the description of theoretical calculation, and added the Supplementary Fig. 41 to better understand the theoretical calculation process.

2. In addition, Figure 3 is titled “Chemical structure of four COFs.” which is not correct. Figure 3 doesn’t give any information for chemical structure, and instead it is about electronic characterizations.

We appreciate this comment. We removed this inaccurate expression. Then, we revised the Figure 3 title as Electronic characterizations of four COFs in the manuscript.

Figure 3. Electronic characterizations of four COFs. The XPS spectra of (A) Co 2p and (B) N 1s for CoTAPP-PATA-COF, N⁺-COF, NH-COF and N⁺-NH-COF. (C) the UV-vis absorption (insert: Tauc plots) and (D) the energy gap (HOMO and LUMO) for CoTAPP-PATA-COF (black), N⁺-COF (green), NH-COF (blue) and N⁺-NH-COF (red).

=====

We appreciate this reviewer very much for the above suggestive comments that greatly improved the quality of the manuscript.

REVIEWERS' COMMENTS

Reviewer #3 (Remarks to the Author):

The authors have nicely addressed my concerns. I would like to recommend its publication.

Reply to Referees' comments

Reply to Referee #3

Reviewer #3: The authors have nicely addressed my concerns. I would like to recommend its publication.

=====

Thank you for your positive comments. We appreciate this reviewer very much for the above suggestive comments that greatly improved the quality of the manuscript.

=====